# Hepatic protein tyrosine phosphatase receptor gamma links obesity-induced inflammation to insulin resistance

Xavier Brenachot[1,2], Giorgio Ramadori[1,2], Rafael M. Ioris[1,2], Christelle Veyrat-Durebex[1,2], Jordi Altirriba [2,3], Ebru Aras[1,2], Sanda Ljubicic[1,2], Daisuke Kohno[4,5], Salvatore Fabbiano[1,2], Sophie Clement[6], Nicolas Goossens[7], Mirko Trajkovski[1,2], Sheila Harroch[8,9], Francesco Negro[2,6,7] & Roberto Coppari[1,2]

Obesity-induced inflammation engenders insulin resistance and type 2 diabetes mellitus (T2DM) but the inflammatory effectors linking obesity to insulin resistance are incompletely understood. Here, we show that hepatic expression of Protein Tyrosine Phosphatase Receptor Gamma (PTPR-γ) is stimulated by inflammation in obese/T2DM mice and positively correlates with indices of inflammation and insulin resistance in humans. NF-κB binds to the promoter of *Ptprg* and is required for inflammation-induced PTPR-γ expression. PTPR-γ loss-of-function lowers glycemia and insulinemia by enhancing insulin-stimulated suppression of endogenous glucose production. These phenotypes are rescued by re-expression of *Ptprg* only in liver of mice lacking *Ptprg* globally. Hepatic PTPR-γ overexpression that mimics levels found in obesity is sufficient to cause severe hepatic and systemic insulin resistance. We propose hepatic PTPR-γ as a link between obesity-induced inflammation and insulin resistance and as potential target for treatment of T2DM.

[1] Department of Cell Physiology and Metabolism, Faculty of Medicine, University of Geneva, 1211 Geneva 4, Switzerland. [2] Diabetes Center of the Faculty of Medicine, University of Geneva, 1211 Geneva 4, Switzerland. [3] Laboratory of Metabolism, Department of Internal Medicine Specialties, Faculty of Medicine, University of Geneva, 1211 Geneva, Switzerland. [4] Advanced Scientific Research Leaders Development Unit, Gunma University, 3-39-15 Showa-machi, Maebashi, Gunma 371-8512, Japan. [5] Metabolic Signal Research Center, Institute for Molecular and Cellular Regulation, Gunma University, 3-39-15 Showa-machi, Maebashi, Gunma 371-8512, Japan. [6] Clinical Pathology, Geneva University Hospitals, Rue Gabrielle Perret-Gentil, 1211 Geneva 14, Switzerland. [7] Gastroenterology and Hepatology, Geneva University Hospitals, Rue Gabrielle Perret-Gentil, 1211 Geneva 14, Switzerland. [8] Department of Psychiatry, New York University Langone School of Medicine, New York, NY 10016, USA. [9] Department of Neuroscience, Institut Pasteur, 75824 Paris, France. Xavier Brenachot and Giorgio Ramadori contributed equally to this work. Correspondence and requests for materials should be addressed to G.R. (email: giorgio.ramadori@unige.ch) or to R.C. (email: roberto.coppari@unige.ch)

Obesity has reached epidemic proportions as 1.9 billion people are currently obese or overweight[1]. One of the most serious complications of obesity is T2DM[2]. Inflammation has been suggested as a mechanistic link between increased adiposity and T2DM[3]. For example, several circulating inflammatory cues increased in obesity have been shown to obstruct insulin action; these include leukotriene B4[4], galectin-3[5], interleukin 1-beta[6], and tumor necrosis factor alpha (TNF-α)[7]. These molecules trigger NF-κB activity which contribute to the development of cellular insulin resistance[3]. Also, JNK, p38, and inflammasome have been suggested to link inflammation to insulin resistance[8–10]. While pharmacological approaches aimed at targeting the aforementioned mechanisms have had encouraging outcomes there is still an unmet need for new, broad, and effective tools for translation into the clinic[11–17]. We reasoned that unmasking the identity of inflammatory mediators linking obesity to insulin resistance could provide new targets for improved T2DM treatment. Here, we show that the expression of hepatic Protein Tyrosine Phosphatase Receptor Gamma (PTPR-γ) is induced by inflammatory signaling and is increased in the context of obesity. While PTPR-γ deletion enhances hepatic insulin sensitivity PTPR-γ overexpression in liver is sufficient to cause hepatic insulin resistance. Thus, our data unveil PTPR-γ as an important link between metabolic inflammation and insulin resistance and a new putative target for anti-T2DM therapy.

## Results

**Hepatic PTPR-γ level correlates with inflammation.** PTPR-γ is upregulated during inflammation[18]. As its role in metabolism is unknown, we first monitored PTPR-γ expression in metabolically relevant tissues of mice either fed on a high-caloric-diet (HCD; an established model of obesity/T2DM)[19, 20] or treated with lipopolysaccharides (LPS; an established model of inflammation/insulin

resistance)[21]. *Ptprg* mRNA level was increased nearly by a factor of two in liver, skeletal muscle, and adipose tissue (AT) of HCD-fed and LPS-treated mice compared to their standard diet (STD)-fed or saline-treated controls, respectively (Fig. 1a, b). In humans, we found that hepatic *PTPRG* mRNA contents increase proportionally to the severity of non-alcoholic fatty liver disease (NAFLD); a condition associated with increased inflammation and insulin resistance[22, 23]. Indeed, subjects affected by non-alcoholic steatohepatitis (NASH; the most severe form of NAFLD) display the highest while subjects affected by NAFL an intermediate level of increased *PTPRG* mRNA content compared to healthy individuals (Fig. 1c). In the same cohort hepatic *PTPRG* expression correlated with the expression of the NF-κB targets *IkBA, IL6, BLIMP-1*, and *CCL28*[24] (Fig. 1d, e and Supplementary Fig. 1a, b). We also directly measured *PTPRG* expression in liver biopsies from another human cohort and found a strong positive correlation between *PTPRG* expression and the expression of NF-κB targets *IkBA, MnSOD*, and *IL6*[24] (Supplementary Fig. 1c–e) and subjects' homeostatic model assessment of insulin resistance (HOMA-IR; an index of insulin resistance), and circulating insulin and glucose levels (Fig. 1f–h). These data demonstrate that in the context of obesity and inflammation PTPR-γ is upregulated in metabolic relevant tissues and suggest that it could play a causative role in insulin resistance in humans.

**PTPR-γ antagonizes hepatic insulin action.** To directly assess the metabolic relevance of PTPR-γ we conducted loss-of-function studies. Mice lacking PTPR-γ (*Ptprg⁻/⁻* mice)[25] and their wild-type littermates (*Ptprg⁺/⁺* mice) were either fed on a HCD or a STD. Chronic feeding on a HCD caused obesity in *Ptprg⁻/⁻* and *Ptprg⁺/⁺* mice as their body weight, fat mass, and circulating level of the adipocytes-secreted hormone leptin were all significantly higher compared to their genetically-matched STD-fed controls;

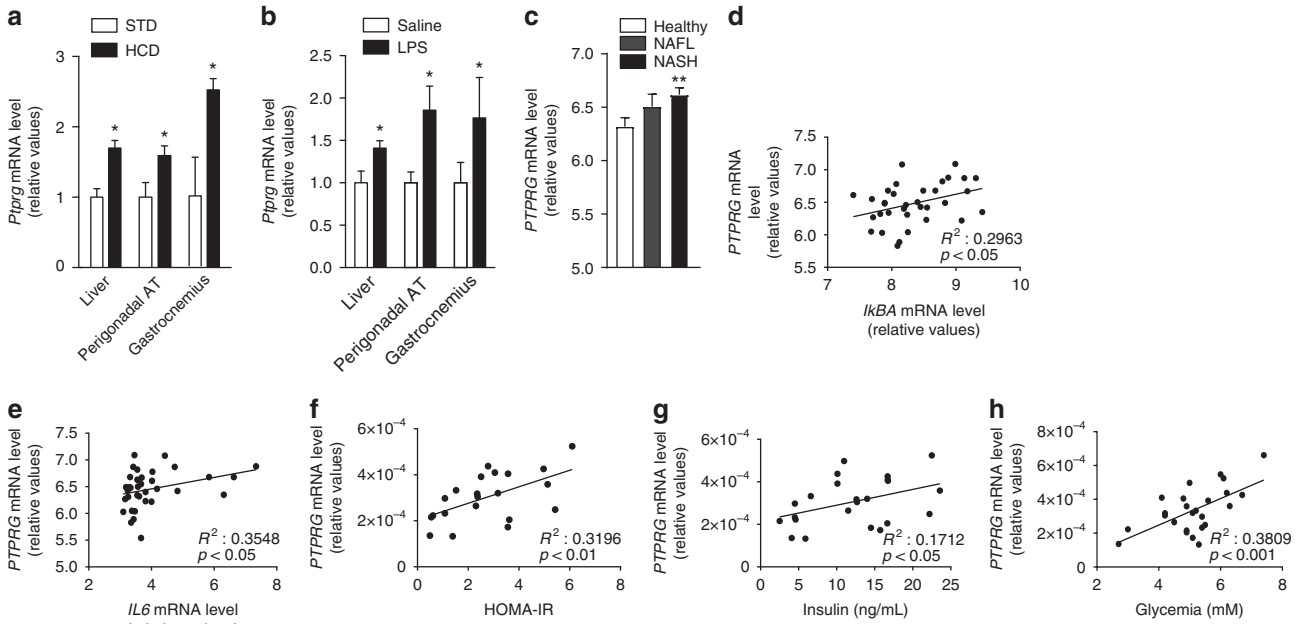

**Fig. 1** Hepatic PTPR-γ expression correlates with inflammation and insulin resistance in mice and humans. **a** Hepatic *Ptprg* mRNA expression level of 30-week-old wild-type mice fed either on a STD or a HCD (*n* = 8 per group). **b** Hepatic *Ptprg* mRNA expression level in mice treated with LPS or saline (*n* = 4 per group). **c** Hepatic *PTPRG* mRNA expression level in patients with non alcoholic fatty liver (NAFL) (*n* = 10) or non-alcoholic steatohepatitis (NASH) (*n* = 16) compared to healthy subjects (*n* = 19) from publicly available microarray E-MEXP-3291. **d–h** Correlation between hepatic *PTPRG* mRNA expression and **d** hepatic *IkBA* mRNA level, **e** hepatic *IL6* mRNA level, **f** HOMA index, **g** plasma insulin level, and **h** glycemia in humans (*n* = 33). Error bars represent SEM. Statistical analyses were done using two-tailed unpaired Student's *t* test. *P < 0.05. Correlation analyses were performed by using the Spearman rank-correlation test. See also Supplementary Fig. 1

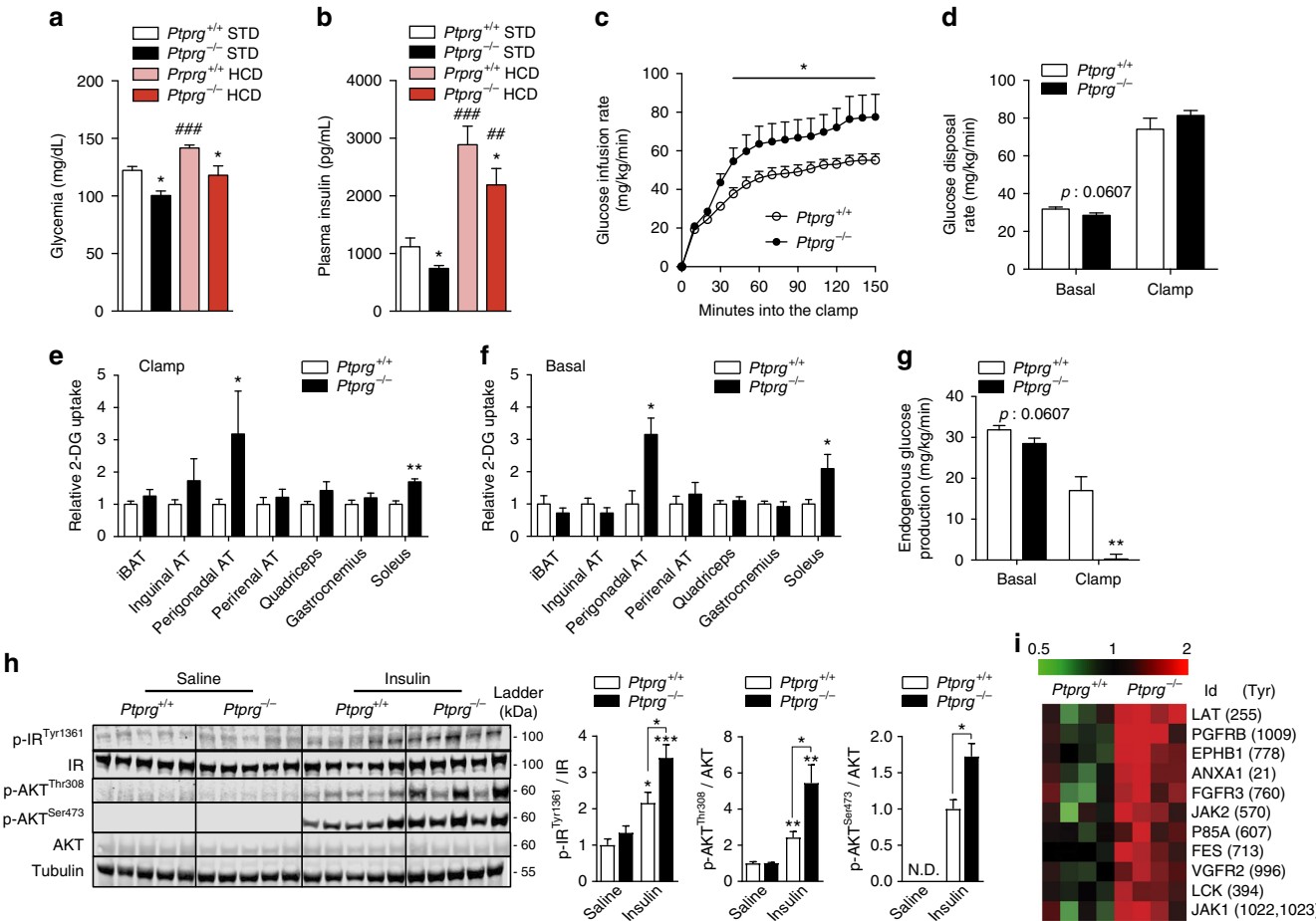

**Fig. 2** Lack of PTPR-γ improves insulin sensitivity. **a** Circulating glucose and **b** insulin level in 30-week-old $Ptprg^{+/+}$ ($n = 8$) and $Ptprg^{-/-}$ ($n = 6$) mice fed either on a STD or a HCD. **c** Glucose infusion rate over time in clamp condition, **d** glucose disposal rate, **e**, **f** peripheral tissues glucose disposal and **g** endogenous glucose appearance rate in basal and clamp condition in 12-week-old $Ptprg^{+/+}$ ($n = 6$) and $Ptprg^{-/-}$ ($n = 6$) mice fed on a STD. **h** Immunoblot of the described total and phosphorylated proteins in the liver of 10-week-old STD-fed $Ptprg^{-/-}$ and $Ptprg^{+/+}$ mice 10 min after an intraperitoneal injection of either insulin (5 units/kg of body weight) or saline. On the right, panels represent relative protein quantification ratios. **i** Hit-map representing phosphorylation at specific tyrosine residues (values are obtained from data shown in Supplementary Table 1 and represent changes between $Ptprg^{-/-}$ and $Ptprg^{+/+}$ mice that are greater than 1.5 fold and of statistical significance ($t$-test: $p$ values < 0.05). Error bars represent SEM. Statistical analyses were done using two-tailed unpaired Student's $t$ test or using one-way ANOVA (Tukey's post test). $^*P < 0.05$, $^{**}P < 0.01$; $^{##}P < 0.01$, $^{###}P < 0.001$. In **a** and **b** the $^*P$ values are calculated by comparing values gathered from mice of different genotype fed on the same diet; $^{##}P < 0.01$, $^{###}P < 0.001$, these values were calculated by comparing data from HCD-fed mice of either genotype vs. STD-fed $Ptprg^{+/+}$ mice. In **h**, $^*P$ values are calculated by comparing values gathered from the same genotype except when differently indicated. (N.D. non-detectable). See also Supplementary Figs. 2, 3, and 4

of note, in both feeding conditions lack of PTPR-γ did not alter these parameters (Supplementary Fig. 2a–c). Caloric intake, energy expenditure, $O_2$ consumption, and $CO_2$ production of $Ptprg^{-/-}$ mice were normal (Supplementary Fig. 2d–g). Also, adipocyte morphology and size distribution and the number of UCP1-expressing brown-like/beige adipocytes and mRNA contents of several brown/beige fat markers were normal in AT of both STD- and HCD-fed $Ptprg^{-/-}$ mice (Supplementary Fig. 2h–j). Thus, PTPR-γ is not required for normal body weight homeostasis.

On the other hand, PTPR-γ deficiency profoundly affected glucose and insulin balance. Chronic feeding on a HCD caused insulin resistance in control mice as glycemia and insulinemia were higher in HCD- compared with STD-fed $Ptprg^{+/+}$ mice; noteworthy, in both feeding contexts circulating glucose and insulin levels were significantly reduced in $Ptprg^{-/-}$ compared with $Ptprg^{+/+}$ mice (Fig. 2a, b). Importantly, the level of circulating glucose in HCD-fed mice lacking PTPR-γ was comparable to the one seen in STD-fed normal mice hence indicating a strong protective role of PTPR-γ deficiency

against development of hyperglycemia (Fig. 2a). $Ptprg^{-/-}$ mice displayed improved glucose tolerance (Supplementary Fig. 3a, b) suggesting that lack of PTPR-γ enhances insulin action. To directly measure insulin sensitivity, we performed hyperinsulinemic/euglycemic clamp and biochemical studies. In keeping with our previous results (Fig. 2a), basal circulating glucose was reduced in $Ptprg^{-/-}$ mice (Supplementary Fig. 3c). During the clamp, circulating insulin levels were experimentally increased and kept similar between groups (Supplementary Fig. 3d). Improved insulin sensitivity was revealed by a higher glucose infusion rate (Fig. 2c) needed to clamp euglycemia (Supplementary Fig. 3e) in $Ptprg^{-/-}$ mice. Although insulin-induced glucose disposal was normal (Fig. 2d), during the clamp the amount of exogenously administered radiolabeled glucose analog 2[14 C]deoxyglucose was increased in AT and skeletal muscle of $Ptprg^{-/-}$ mice (Fig. 2e). To test whether increased glucose disposal in these tissues is associated with changes in insulin action at these sites we performed glucose tracing analysis also in basal condition. Despite the fact that $Ptprg^{-/-}$ mice have reduced circulating insulin level (Fig. 2b), the

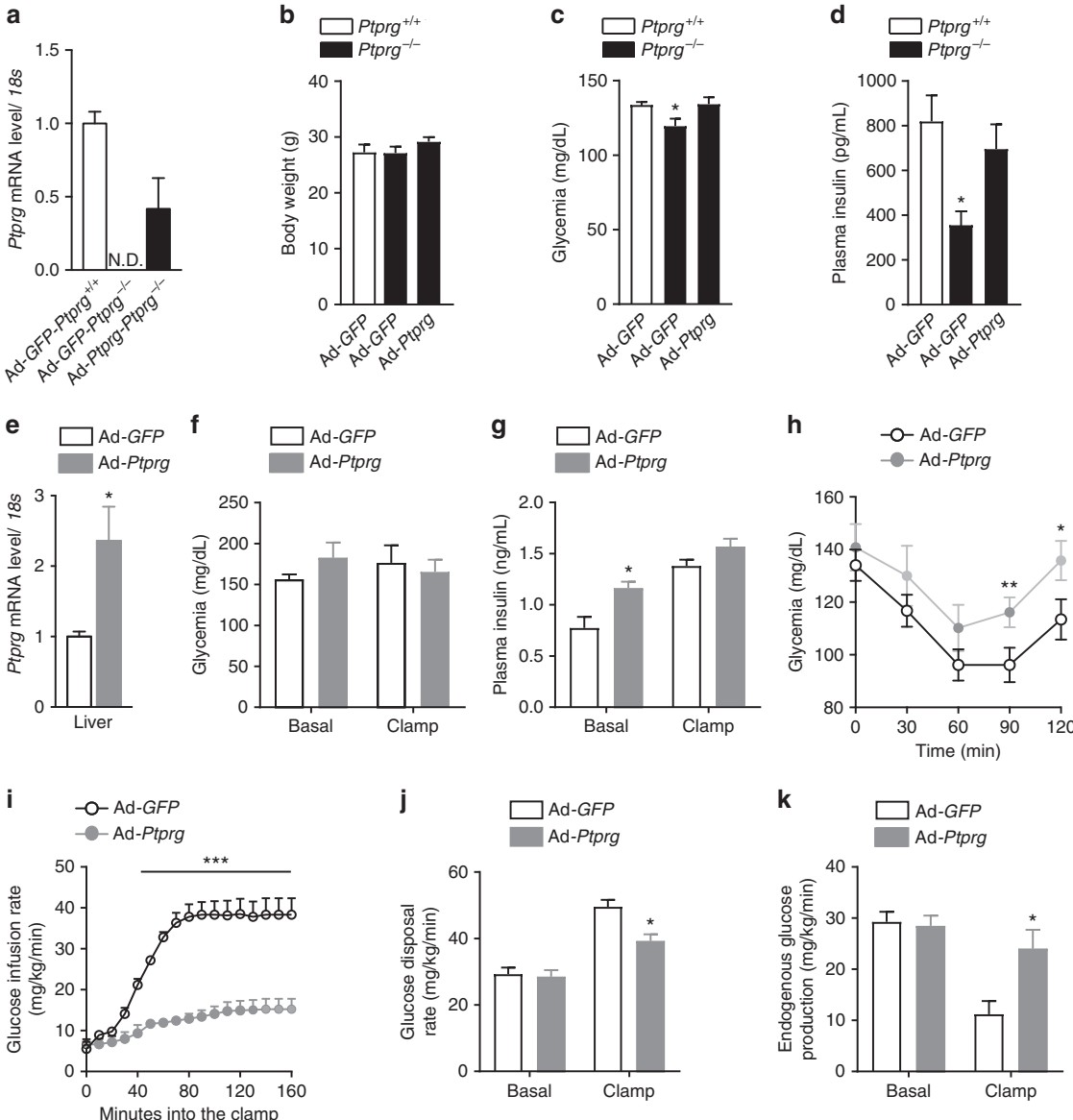

**Fig. 3** Hepatic overexpression of PTPR-γ is sufficient to cause insulin resistance. **a** Hepatic *Ptprg* mRNA expression, **b** body weight, **c** glycemia, and **d** cirualting insulin level of 10-week-old STD-fed *Ptprg*$^{-/-}$ and *Ptprg*$^{+/+}$ mice 10 days after infection with either control adenovirus (Ad-*GFP*) or adenovirus expressing mouse *Ptprg* (Ad-*Ptprg*) (n = 6–9/experimental group). **e–k** Values from 10-week-old FVB mice infected with either control adenovirus (Ad-*GFP*) or adenovirus expressing mouse *Ptprg* (Ad-*Ptprg*). **e** mRNA level from the liver (14 days post infection) (n = 8). **f** Glycemia and **g** insulinemia in basal and clamp condition (10 days post viral infection) (n = 7). **h** Glycemia during an insulin tolerance test (ITT; performed 12 days after viral infection; the dose of insulin used was 0.75 U/kg of body weight) (n = 7). **i** Glucose infusion rate over time, **j** glucose disposal rate, and **k** endogenous glucose appearance rate in basal and clamp condition (n = 7). Error bars represent SEM. Statistical analyses were done using two-tailed unpaired Student's *t* test or using one-way ANOVA (Tukey's post test). *P < 0.05; **P < 0.01, ***P < 0.001. See also Supplementary Fig. 5

amounts of radiolabeled 2[14 C]deoxyglucose in AT and skeletal muscle of *Ptprg*$^{-/-}$ mice was increased also in basal condition (Fig. 2f). Next, to independently assess insulin sensitivity in these tissues we measured the statuses of phosphorylated insulin receptor (IR) and AKT (which have been widely used as measures of insulin action)[26]. Insulin administration increased phosphorylation status of IR and AKT in AT and soleus skeletal muscle of control mice (Supplementary Fig. 3f, g). In keeping with the basal glucose tracing results (Fig. 2f) and the fact that whole-body insulin-induced glucose disposal was normal (Fig. 2d), insulin-stimulated phosphorylation of IR and AKT in AT and soleus skeletal muscle of *Ptprg*$^{-/-}$ mice was similar to *Ptprg*$^{+/+}$ mice (Supplementary Fig. 3f, g). These data suggest that the enhanced glucose uptake in AT and skeletal muscle of mice

lacking PTPR-γ is independent to changes in insulin sensitivity at these sites.

Remarkably, the ability of insulin to suppress endogenous glucose production (EGP) was prominently augmented in *Ptprg*$^{-/-}$ mice as during the clamp this parameter was ~98% lower in mice lacking PTPR-γ compared with controls (Fig. 2g). As the liver largely contributes to EGP[27], we independently assessed hepatic insulin sensitivity. Insulin administration increased phosphorylation status of IR and AKT in liver of control mice (Fig. 2h and Supplementary Fig. 3h). In accordance with the clamp results (Fig. 2g), insulin-stimulated phosphorylation of IR and AKT was enhanced in liver of STD-fed *Ptprg*$^{-/-}$ compared with *Ptprg*$^{+/+}$ mice (Fig. 2h). Similarly, we found increased hepatic insulin signaling in *Ptprg*$^{-/-}$ mice in the context

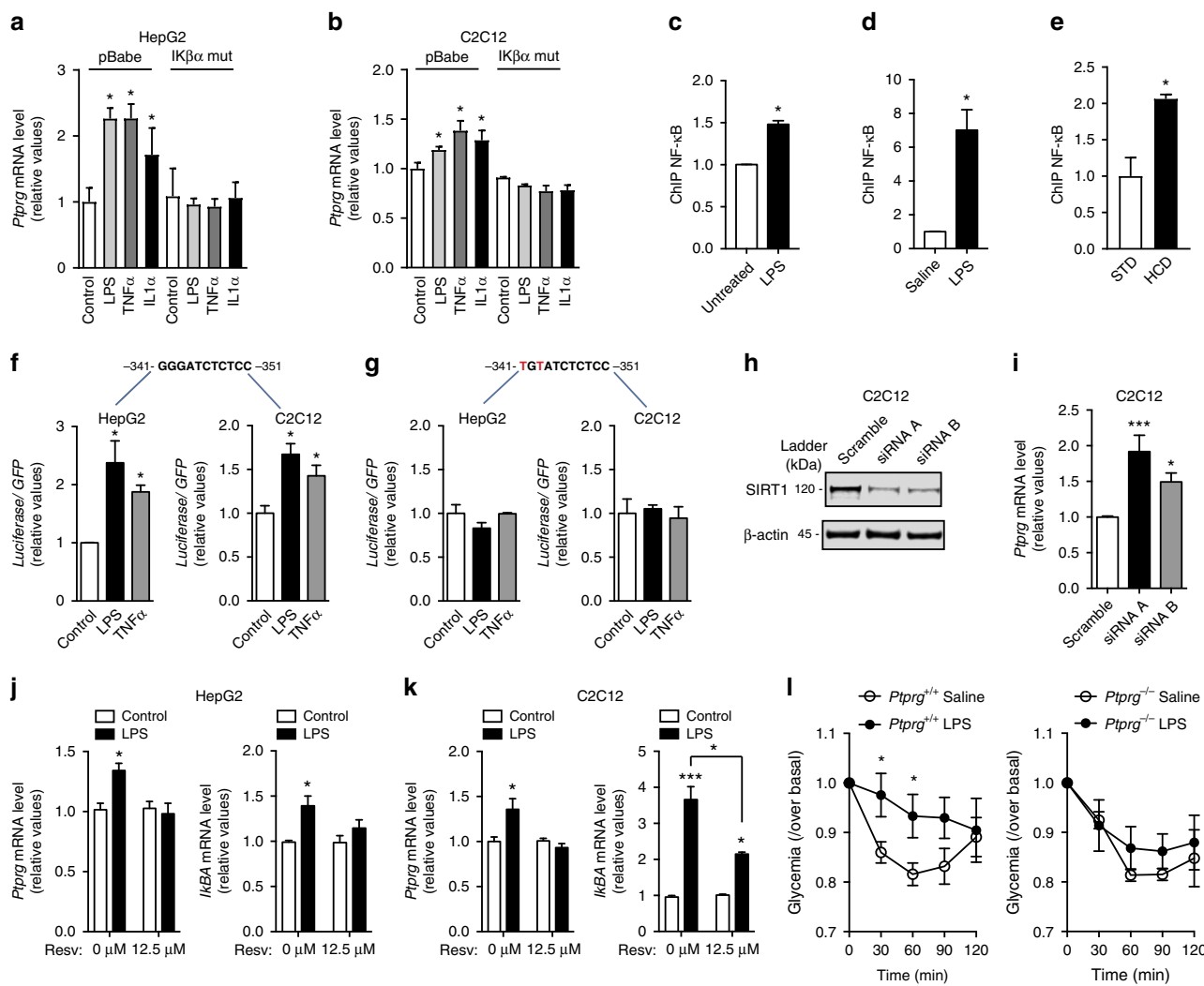

**Fig. 4** NF-κB mediates inflammation-induced PTPR-γ expression. **a** *Ptprg* mRNA level in HepG2 cells and **b** C2C12 cells treated as indicated. (IκBα DN expresses a mutant "super-repressor" allele of IκBα harboring two amino acid substitutions: Serine 32 and 36 are substituted in Alanine). **c** ChIP analyses of NF-κB in C2C12 cells treated with LPS or vehicle (relative data are represented over IgG controls). **d, e** ChIP analyses of NF-κB in liver of mice treated with saline or LPS and STD or HCD (relative data are represented over IgG controls). **f, g** mRNA levels of Firefly luciferase in HepG2 and C2C12 cells transduced with pGL2-basic plasmid containing *Ptprg* promoter region containing a mutated/disrupted NF-κB putative binding motif and treated as indicated. **h** Knockdown in HepG2 cells of SIRT1 using two different siRNAs and **i** *Ptprg* mRNA level in the same cells. **j, k** *Ptprg* mRNA level in HepG2 and C2C12 cells treated with resveratrol as indicated. **l** Insulin tolerance test (ITT) in 12-week-old mice treated 28 days with LPS (the dose of insulin injected intra-peritoneally is 0.75 U/kg). Error bars represent SEM. Statistical analyses were done using two-tailed unpaired Student's *t* test or using one-way ANOVA (Tukey's post test). $^*P < 0.05$, $^{**}P < 0.01$, $^{***}P < 0.001$. See also Supplementary Fig. 6

of HCD feeding (Supplementary Fig. 4a). Also, insulin-stimulated phosphorylation of these proteins was increased in primary hepatocytes obtained from *Ptprg*$^{-/-}$ compared to *Ptprg*$^{+/+}$ mice (Supplementary Fig. 4b) suggesting a cell-autonomous effect of PTPR-γ in antagonizing insulin action in hepatocytes. To gain more mechanistic insights into the function of PTPR-γ, we measured the activity of phosphatases/kinases in lysates obtained from liver samples of mice lacking PTPR-γ and their controls. By using arrays with immobilized tyrosine peptide substrates we were able to identify known (e.g., JAK2)[28] and putative new targets of PTPR-γ. Our new results shown in Fig. 2i and Supplementary Table 1 combined with our previous data shown in Fig. 2g, h and Supplementary Fig. 3a indicate that PTPR-γ affects hepatic insulin signaling and glucose metabolism by altering phosphorylation status of proteins whose metabolic role is known (e.g., IR, P85A, AKT) and also of others whose function on hepatic insulin signaling and glucose metabolism is inadequately understood (e.g., LAT, PGFRB, EPHB1, ANXA1, FGFR3,

JAK2, FES, VGFR2, LCK, and JAK1). Together, our data demonstrate that lack of PTPR-γ greatly improves glucose homeostasis at least in part by enhancing hepatic insulin sensitivity in a body-weight-independent manner.

**Hepatic PTPR-γ overexpression causes insulin resistance.** To determine the tissue underlying the glucoregulatory action of PTPR-γ we re-expressed PTPR-γ in liver of *Ptprg*$^{-/-}$ mice. To this end, we generated recombinant adenoviruses with high degree of hepatotropism (serotype 5) expressing either PTPR-γ and GFP concomitantly (Ad-*Ptprg*-IRES-*GFP*, henceforward referred to as Ad-*Ptprg*) or only GFP (Ad-*GFP*). As expected, GFP mRNA was detected in liver but not in AT and skeletal muscle of Ad-*Ptprg*- and Ad-*GFP*-injected mice (Supplementary Fig. 5a). While hepatic *Ptprg* mRNA was undetectable in Ad-*GFP Ptprg*$^{-/-}$ mice it was expressed in Ad-*Ptprg Ptprg*$^{-/-}$ mice (Fig. 3a). In keeping with our previous results, Ad-*GFP Ptprg*$^{-/-}$ mice had reduced glycemia and insulinemia (and normal body weight); however,

these phenotypes were rescued by expression of PTPR-γ only in the liver as Ad-*Ptprg Ptprg*[−/−] mice displayed normal glycemia and insulinemia (and body weight) (Fig. 3b–d).

Next, we set out to determine whether the pathological overexpression of PTPR-γ in the liver of LPS-treated or HCD-fed rodents (Fig. 1a, b) and humans (Fig. 1c–g) contributes to their insulin resistance. To this end, we delivered Ad-*Ptprg* or Ad-*GFP* in wild-type mice and generated a model overexpressing hepatic PTPR-γ at levels similar to the ones seen in insulin resistance. Indeed, Ad-*Ptprg* mice displayed hepatic *Ptprg* mRNA content increased by a factor of two compared to Ad-*GFP* controls (Fig. 3e); this level of overexpression resembles to the one observed in HCD-fed obese and T2DM mice (Fig. 1a). Ad-*Ptprg* mice had a tendency to hyperglycemia, hyperinsulinemia, and reduced responsiveness to the glucose-lowering action of injected insulin (Fig. 3f–h) suggesting that pathophysiological overexpression of hepatic PTPR-γ is sufficient to cause insulin resistance. To directly measure insulin sensitivity, we performed hyperinsulinemic/euglycemic clamp and biochemical studies. During the clamp, circulating insulin levels were experimentally increased and kept similar between groups (Fig. 3g). Remarkable impairment in insulin sensitivity was revealed by a 4-fold reduction in glucose infusion rate (Fig. 3i) needed to clamp euglycemia (Fig. 3f) in Ad-*Ptprg* mice. Insulin-induced glucose disposal was decreased in Ad-*Ptprg* mice (Fig. 3j). To determine the tissue(s) underlying this defect we measured the amounts of exogenously administered radiolabeled 2[14 C]deoxyglucose. While glucose uptake in skeletal muscle was normal it was reduced in AT of Ad-*Ptprg* mice (Supplementary Fig. 5b). The test whether decreased glucose disposal in AT is dependent to changes insulin action in this tissue we assessed insulin sensitivity biochemically. Insulin administration increased phosphorylation status of IR and AKT similarly in AT of Ad-*Ptprg* and control mice (Supplementary Fig. 5c). These data suggest that in mice overexpressing hepatic PTPR-γ the diminished glucose uptake in AT is independent of changes in insulin signaling at this site.

While PTPR-γ deficiency greatly enhanced the ability of insulin to suppress EGP (Fig. 2g) hepatic PTPR-γ overexpression significantly impaired insulin action on the liver (Fig. 3k). In line with the clamp data, insulin-induced phosphorylation of IR and AKT was reduced in liver of Ad-*Ptprg* mice (Supplementary Fig. 5d). The impaired insulin sensitivity caused by hepatic PTPR-γ overexpression was not due to changes in body weight as this parameter was normal in Ad-*Ptprg* mice (Supplementary Fig. 5e). Collectively, these data demonstrate that hepatic PTPR-γ overexpression at a level similar to one observed in obesity is sufficient to cause hepatic insulin resistance.

## Ptprg expression is induced by NF-κB and suppressed by SIRT1

Our results demonstrate that hepatic *Ptprg* expression is induced in the context of obesity/T2DM and inflammation and that it greatly contributes to the development of insulin resistance. Therefore, understanding the regulatory mechanism of *Ptprg* expression is of pathophysiological relevance. Obesity and T2DM are characterized by low-grade inflammation[7, 29]. Indeed, the circulating levels of pro-inflammatory cues TNF-α and IL-6, as well as the mRNA levels of the NF-κB target IκBα in liver, skeletal muscle, and AT were all increased in HCD-fed *Ptprg*[−/−] and *Ptprg*[+/+] mice compared to their genetically matched STD-fed controls (Supplementary Fig. 6a, b). Of note, in both feeding conditions lack of PTPR-γ did not alter these parameters (Supplementary Fig. 6a, b). The levels of several inflammation markers in plasma, liver, skeletal muscle, and AT were also similar betweeen Ad-*Ptprg* and Ad-*GFP* mice (Supplementary Fig. 6c–e).

These data suggest that the insulin sensitizing or inhibiting action of PTPR-γ deficiency or overexpression, respectively, is not due to changes in pro-inflammatory inputs.

To directly determine whether increased PTPR-γ expression in HCD-fed or LPS-treated mice is due to inflammation or secondary to the many systemic changes brought about by HCD feeding or LPS treatment we performed in vitro experiments. In two unrelated cell lines (i.e., hepatic HepG2 and muscle C2C12 cells) treatment with the inflammatory cues LPS, TNF-α, or interleukin-1 alpha (IL-1α) stimulated expression of *MnSOD* (Supplementary Fig. 6f) and also of *Ptprg* (Fig. 4a, b); thus, inflammation promotes *Ptprg* expression directly. To identify the mechanism(s) underlying inflammation-driven *Ptprg* expression we focused on NF-κB because we observed a κB binding site in the promoter of *Ptprg* (Supplementary Fig. 6g). Thus, by performing chromatin immunoprecipitation (ChIP) assays we investigated whether NF-κB occupies the promoter region of *Ptprg* and whether enhanced inflammatory signaling augments this occupancy. Our data indicate that NF-κB occupies the promoter region of *Ptprg* and that this occupancy is increased by LPS treatment in vitro (Fig. 4c) and in vivo (Fig. 4d). Also NF-κB occupancy of *Ptprg* promoter is increased by HCD feeding in vivo. (Fig. 4e). To determine whether NF-κB is required for inflammation-induced *Ptprg* expression we inhibited NF-κB activity by expressing an established mutant super-repressor allele of IκBα[30]. In hepatic HepG2 and muscle C2C12 cells, expression of mutant IκBα (IκBα DN) abolished LPS-, TNF-α-, or IL-1α-induced overexpression of *MnSOD* (Supplementary Fig. 6f) and also of *Ptprg* (Fig. 4a, b). Thus, inflammation is sufficient to induce expression of *Ptprg* and NF-κB is required for this effect in vitro. To determine whether *Ptprg* promoter region is sufficient for inflammation-induced gene expression, we cloned this region containing the κB recognition site upstream of a luciferase reporter cassette and generated a reporter vector (Supplementary Fig. 6h). This promoter sequence was sufficient for LPS- or TNF-α-induced gene expression as treatment with these cues enhanced luciferase mRNA content in cells transformed with the reporter vector (Fig. 4f). Also, we mutated this vector as such the κB recognition site is changed from 5′-GGGATCTCTCC-3′ to 5′-TGTATCTCTCC-3′ (Supplementary Fig. 6h); this mutation is known to abolish NF-κB binding[31]. Of note, LPS or TNF-α treatment was no longer able to enhance luciferase mRNA content in cells transformed with the mutant vector (Fig. 4g). Collectively these results demonstrate that *Ptprg* expression is induced by inflammation via activation of NF-κB.

SIRT1 is a nicotinamide adenine dinucleotide (NAD[+])-dependent protein deacetylase[32] that has been reported to protect against development of metabolic imbalance and to mediate, at least in part, the metabolic beneficial effects of resveratrol administration[33−38]. Because SIRT1 inhibits NF-κB activity[39], we tested whether SIRT1 antagonizes the action of NF-κB on *Ptprg* expression. First, we found that SIRT1 down-regulation is sufficient to cause increased *Ptprg* expression in vitro and in vivo (Fig. 4h, i and Supplementary Fig. 6i). Second, our results indicate that treatment with the SIRT1 activator resveratrol dampens the stimulatory action of LPS on the expression of IKβα and *Ptprg* (Fig. 4j, k). Moreover, hypercaloric-fed mice treated with the NAD[+] precursor and SIRT1 activator nicotinamide riboside[40] displayed reduced expression of hepatic IKβα, *MnSOD*, and *Ptprg* compared to their untreated controls (Supplementary Fig. 6j). These results suggest that *Ptprg* expression is induced by inflammation via activation of NF-κB and indicate that this effect might be antagonized by SIRT1 activation.

To further test whether PTPR-γ is required for inflammation-induced insulin resistance we compared the effect of LPS treatment on insulin sensitivity in mice lacking PTPR-γ and their intact

controls. LPS treatment led to similar increases in expression level of hepatic IKβα and *MnSOD* and plasma content of TNF-α and IL-6 between *Ptprg*[−/−] and *Ptprg*[+/+] mice (Supplementary Fig. 6k–n). As previously described[41], LPS treatment caused insulin resistance; however, *Ptprg*[−/−] mice were protected from LPS-induced insulin resistance (Fig. 4i). Thus, our data indicate that LPS-induced insulin resistance requires, at least in part, PTPR-γ.

## Discussion

Here we show that PTPR-γ is a negative regulator of hepatic insulin signaling in both physiological and obesity/inflammation contexts. Importantly, our results suggest a model in which PTPR-γ links hepatic inflammation with insulin resistance and glucose imbalance associated with obesity. Because PTPR-γ loss- or gain-of-function does not alter body weight we suggest that PTPR-γ regulates insulin signaling directly. Our results indicate that the liver is a major site for the glucoregulatory action of PTPR-γ and provide a mechanistic explanation for previous findings showing that genetically-mediated induction of hepatic NF-κB activity in lean mice causes insulin resistance[20], while hepatic-specific inactivation of NF-κB prevents insulin resistance[41]. As an NF-κB target whose overexpression in liver is sufficient to cause insulin resistance in lean mice and whose hepatic level correlates with indices of inflammation and insulin resistance in humans, our data also indicate that hepatic PTPR-γ is an important molecular component of inflammation-driven insulin resistance in rodents and humans.

Several questions arise from the current study. For example, in mice lacking PTPR-γ insulin action in AT and skeletal muscle is normal (Supplementary Fig. 3f, g); yet, glucose uptake in these tissues is elevated (Fig. 2e, f). These results suggest that PTPR-γ in extra-hepatic tissues may also be metabolically relevant. This issue could be tackled by future studies in mice lacking PTPR-γ in a tissue-specific manner. Also, does PTPR-γ regulate hepatic glucose metabolism by targeting IR and/or other substrates? Interestingly, IR[42] and JAK2[28] have been shown to be dephosphorylated by PTPR-γ and our data are in line with these results (Fig. 2h and Supplementary Figs. 3h and 4b). Also, we identified other putative PTPR-γ targets (Fig. 2i and Supplementary Table 1). Experiments aimed at testing their role in mediating the effects of PTPR-γ on glucose metabolism in vivo are warranted. Furthermore, are the beneficial effects of SIRT1 activators (e.g., resveratrol, nicotinamide riboside) mediated by suppression of hepatic PTPRγ? While our in vitro and in vivo data shown in Fig. 4h–k and Supplementary Fig. 6i, j would suggest that this may be the case future experiments in resveratrol- or nicotinamide-riboside-treated diabetic mice with our without PTPRγ are needed.

Our data suggest that means aimed at decreasing hepatic PTPR-γ expression/activity should improve T2DM. This goal could be achieved by development of specific PTPR-γ inhibitors. For example, the improved insulin action shown in mice lacking Protein Tyrosine Phosphatase 1B (PTP1B) propelled efforts aimed at generating PTP1B inhibitors[43, 44]. Because glucose/insulin phenotypes displayed by mice lacking PTPR-γ are comparable, or even superior, to the ones of PTP1B null mice[43] it is possible that our results trigger interests for developing PTPR-γ inhibitors. Also, based on the available PTPR-γ crystal structure data another approach could be pursued. PTPR-γ is thought to exist in two conformations: the active monomer and the inactive homodimer brought about by PTPR-γ binding to its endogenous ligand(s)[42]. Several members of the contactin family have been suggested as endogenous PTPR-γ ligands[45, 46]. Hence, experiments aimed at testing whether these ligands inhibit PTPR-γ[42, 47] and improve T2DM are warranted.

In summary, our results indicate that the increased hepatic PTPR-γ level observed in obesity is sufficient to cause insulin resistance and hence unveil PTPR-γ as a new target for anti-T2DM therapy.

## Methods

**Animal studies.** The *Ptprg* null allele was generated, as previously described[25]. By breeding *Ptprg*[−/+] males with *Ptprg*[−/+] females we obtained *Ptprg* null (*Ptprg*[−/−]) males and females and their wild-type (*Ptprg*[+/+]) controls. Only male littermates were randomly assigned to experimental groups and compared with each other. Recombinant adenoviruses serotype 5 expressing either PTPR-γ and GFP concomitantly (Ad-*Ptprg*) or only GFP (Ad-*GFP*) were generated by VectorBiolabs (USA). *Ptprg*[−/−] mice re-expressing PTPR-γ in liver and their controls were generated as follows: 10-week-old *Ptprg*[−/−] and *Ptprg*[+/+] mice were injected with 10[9] PFU into tail vein in 200 μL of saline. Hepatic PTPR-γ overexpression was achieved by injecting 10[9] PFU into tail vein of 10-week-old FVB mice in 200 μL of saline. Mice were housed in groups of 3–5 with food, either a standard chow rodent diet (RM3, SDS, Essex, UK, Kcal content: 3.61 Kcal/g) or a high-fat diet (HCD; D12331 from Research Diets, New Brunswick, NJ, USA. Kcal content: 5.56 Kcal/g), and water available ad libitum in light- and temperature-controlled environments unless otherwise specified. HCD-fed mice were fed on a standard chow diet up to 8 weeks of age and then switched and maintained on a HCD diet. LPS and saline-treated mice were generated as follows: 8-week-old C57BL/6 J males were administered with either saline or LPS (300 μg/kg/day, Lipopolysaccharides from *Escherichia coli* 0127:B8, L3129, Sigma-Aldrich) for 28 days using osmotic pumps (model 2004; Alzet) inserted subcutaneously into the back of animals. Care of mice at University of Geneva was within the procedures approved by animal care and experimentation authorities of the Canton of Geneva, Switzerland (animal protocol numbers GE/22/15, GE/28/13, GE/120/15). The number of mice used in each experiment is mentioned in the corresponding figure legends.

**Human studies.** For this retrospective analysis, we included RNA extracted from liver biopsy samples taken for diagnostic purposes from 20 male and 13 female patients who were 25–69 years old (average age was 44-year-old) with histologically confirmed chronic hepatitis C who were seen at the Division of Gastroenterology and Hepatology of the University Hospital of Geneva. In total, 18 of these samples were obtained from a multicenter treatment trial[48] and 15 were obtained from a study evaluating molecular correlates of steatosis. Both studies were approved by the institutional review board of the Hospital University of Geneva (HUG) (ethical approval obtained by HUG: reference number #02-070) and a written informed consent was obtained from all patients. Total RNA from liver samples was extracted using the AllPrep DNA/RNA Mini Kit (Qiagen). The quality of the RNA was assessed using an Agilent 2100 Bioanalyser (Agilent Technologies). For all patients, histological (e.g., steatosis and activity scores) and biochemical data (plasma glucose and serum insulin) were gathered. The homeostasis model assessment of insulin resistance (HOMA-IR) was calculated as fasting insulin (mIU L[−1])×fasting glucose (mmol L[−1])/22.5.

**Cell lines and primary cultures.** HepG2 and C2C12 cells lines were cultured accordingly to the repository specifications (ATCC) and treated as follows. Specifically, C2C12 and HepG2 were maintained in DMEM (4.5 g/L glucose, 110 mg/L pyruvate, 4 mM L-glutamine, 100 U/mL Pen/Strep) supplemented with 10% fetal bovine serum (PanBiotech). Experiments were done in 6 well-plate with 50,000 cells plated and cells were treated for 1 day with 0.1 ng TNF alpha (Thermofisher), 1 μg LPS (Sigma) or 1 h with 0.1 ng IL-1 alpha (Thermofisher). Primary hepatocytes from age-matched *Ptprg*[−/−] and *Ptprg*[+/+] mice were established and cultured following an established protocol. Briefly, KREBS-EGTA buffer was perfused in the portal vein and cavea vein incised to evacuate washing buffer for 10 min. Liver was digested with KREBS containing liberase (Roche) for 10 supplementary minutes. Liver was removed and hepatocytes carefully isolated by centrifugation and plated on collagen treated 6 well-plate with a density of 1,000,000 cells/well. They were maintained 16 h in Williams' E Medium (2 g/L glucose, 25 mg/L pyruvate, 2 mM L-glutamine, 100 U/mL Pen/Strep) supplemented with 10% fetal bovine serum and serum starvation was carried out for 3 h in Williams' E Medium without FBS and Pen/Strep. Primary hepatocytes were treated with insulin (Sigma) for 10 min and proteins were quickly extracted with RIPA buffer. All cells were cultured in a humidified incubator at 37 °C and 5% CO$_2$. All in vitro experiments were repeated three times.

**Assessment of mRNA and protein content.** Mice were sacrificed, tissues quickly removed, snap-frozen in liquid nitrogen and subsequently stored at −80 °C. RNAs were extracted using Trizol reagent (Invitrogen). Complementary DNA was generated by Superscript II (Invitrogen) and used with SYBR Green PCR master mix (Applied Biosystem, Foster City, CA, USA) for quantitative real time PCR (q-RTPCR) analysis. mRNA contents were normalized to *18s* mRNA levels. All assays were performed using an Applied Biosystems QuantStudio® 5 Real-Time PCR System. For each mRNA assessment, q-RTPCR analyses were repeated at least three times. Proteins were extracted by homogenizing samples in lysis buffer (Tris 20 mM, EDTA 5 mM, NP40 1% (v/v), protease inhibitors (P2714-1BTL from Sigma, St. Louis, MO, USA), then resolved by SDS–PAGE and finally transferred to a nitrocellulose membrane by electroblotting. Insulin-induced phosphorylation of

proteins and basal proteins levels were assessed by using commercially available antisera as previously described[49, 50]. Briefly, nitrocellulose membranes were blocked with Odissey blocking buffer (Li-Cor Biosciences) 1 h at room temperature and successively incubated 12 h at 4 degrees Celsius with the specific primary antibodies diluted 1:1000 in PBS-T buffer. PBS-T buffer is PBS buffer (27.6 g Sodium Phosphate, dibasic + 160 g Sodium Chloride in 20 L distilled $H_2O$ at pH 7.4) and 0.5% Tween-20. Detection was obtained by using near-infrared secondary antibodies diluted 1:5000 in PBS-T (IRDye 800CW and IRDye600RD; Li-Cor Biosciences) and a Clx Odissey infrared scanner (Li-Cor Biosciences). The list of antibodies and the primer sequences used for q-RTPCR analyses are shown in Supplementary Table 2.

**Blood chemistry.** Fed hormones/metabolites levels were determined by collecting tail blood from mice that were without food for 3 h. Fasted hormones/metabolites levels were assessed in mice provided only with water ad libitum and without food for the indicated period. Time at day at which blood was collected was the same between groups. Tail vein blood was assayed for glucose levels using a standard glucometer (Nova Biomedical). Plasma was collected by centrifugation in EDTA-coated tubes (Kent Scientific) and assayed for leptin (Crystal Chem. Inc., Downers Grove, IL), insulin (Crystal Chem. Inc.), TNF-α (Biovision) and IL-6 (Biovision) levels using the indicated commercially available kits.

**Chromatin immunoprecipitation assays.** ChIP assays were performed as previously described[51] with a few modifications. Cross-linking was obtained with 1% paraformaldehyde for 20 min at room temperature. The reaction was quenched for 10 min at room temperature by adding 0.125 M glycine. After three washes with 1× PBS, cells were lysed with lysis buffer (1% SDS, 10 mM EDTA pH 8, 50 mM Tris-HCl pH 8) supplemented with protease and deacetylase (TSA) inhibitors. Lysates were sonicated on ice using a Bioruptor sonicator (2 pulses of 10 min, 0.5 min sonication). Size of fragments obtained (between 200 and 1200 bp) was confirmed by electrophoresis. Soluble chromatin was collected after centrifugation at 14,000 rpm at 4 °C for 10 min. Soluble chromatin was precleared with 100 µg/mL of salmon sperm DNA (Amersham Biosciences), 2.5 µg/mL of unspecific IgGs, and protein-A-Sepharose at 50% overnight at 4 °C in rotation. After centrifugation (3000 rpm, 3 min at 4 °C), supernatants were collected and specific antibody was added. The antibody used was 5 µL anti-NF-κB p65 (Cell Signaling #8242). A control was performed with unspecific IgGs (AbCam). Also, 1–5% of soluble precleared chromatin was kept at −20 °C as Inputs. Mixtures were incubated at 4 °C for 6 h in rotation and then incubated overnight at 4 °C in rotation with protein-A-Sepharose at 50% (Roche). Beads were collected and washed sequentially at 4 °C for 10 min with TSE I (150 mM NaCl, 0.1% SDS, 1% Triton X-100, 2 mM EDTA, and 20 mM Tris-HCl (pH 8.1)), TSE II (500 mM NaCl, 0.1% SDS, 1% Triton X-100, 2 mM EDTA, and 20 mM Tris-HCl (pH 8.1)), and buffer III (0.25 LiCl, 1% Nonidet P−40, 1% deoxycholate, 1 mM EDTA, and 10 mM Tris−Hcl (pH 8.1)). Beads were washed once with 1× PBS by pipetting and immunoprecipitates were eluted two times (20-min incubation) with elution buffer (0.1 M NaHCO3 and 1% SDS). Reversion of cross-linking was performed overnight by heating samples and input controls at 65 °C, and DNA was purified using the QIAquick spin kit (Qiagen). Real time PCR was performed, as described above. Data were normalized to values obtained with unspecific IgGs (Abcam). The primer sequences used for ChIP analysis are shown in Supplementary Table 2.

**Assessment of energy homeostasis.** Body fat and lean mass were determined using the EchoMRI-700 system and caloric intake, energy expenditure, $O_2$ consumption, and $CO_2$ production were measured using the TSE labmaster system available in the core facility of the Faculty of Medicine of the University of Geneva (http://www.unige.ch/medecine/en/recherche/corefacilities/).

**Immunohistochemistry analyses.** Hematoxylin and eosin staining and UCP1 immunohistochemistry were performed using paraffin-embedded adipose tissue sections as previously described[52]. Briefly, 6 µm-thick sections were mounted on glass slides and treated for paraffin removal by serial passages (each of them 5′) in xylene and then PBS containig decreasing concentration of ethanol (100%, 70%, 50%, and 0%). All steps were performed at room temperature on an orbital shaker at ∼50 rpm. Sections stored in PBS were first washed in PBS three times for 10 min each. Sections were then incubated in 0.3% $H_2O_2$ for 30 min to block the endogenous peroxidase activity and subsequently rinsed in PBS three times for 10 min each. All sections were then placed in blocking solution of 3% bovine serum albumin in PBT-Azide (2.5 mL of Triton X-100 in 1000 mL of PBS-Azide) for 2 h. The sections were then transferred into 1:5000 primary antisera (see Supplementary Table 2) with 3% bovine serum albumin in PBT-Azide overnight. The sections were then rinsed with PBS six times for 10 min each. Next, sections were transferred into 1:1000 biotin-conjugated donkey anti-rabbit secondary antisera (Jackson Immuno Research) with 3% normal donkey serum in PBT for 2 h and subsequently rinsed in PBS three times for 10 min each. Avidin Biotin Complex (Vectastain Elite PK-6100 ABC kit, Vector Laboratories) solution was prepared (1:500 in PBS) 30 min prior to incubating the sections in it for 1 h and subsequently rinsed in PBS two times for 10 min each. The sections were then incubated in 0.04% 3,3′-Diaminobenzidine and 0.01% $H_2O_2$ in PBS for 8 min. Finally, the sections were washed in PBS two times for 10 min each and mounted on gelatin coated glass slides for visualization. Adjacent sections were used for hematoxylin and eosin (H&E) staining.

**Assessment of glucose homeostasis.** Hyperinsulinemic-euglycemic clamps were performed in age-matched conscious unrestrained catheterized mice as previously described[53]. Briefly, catheters were surgically implanted 7 days prior to the experiment, in the right jugular vein and exteriorized above the neck using vascular access button (Instech Laboratories Inc., Plymouth Meeting, PA). Mice were fasted for 3 h, followed by a 2-h infusion of [3-$^3$H] glucose (0.05 µCi/min) (Perkin Elmer, Walthman, MA). Continuous insulin infusion (1.5 mIU/kg body weight/min, Umuline®, Lilly France, Neuilly-sur-Seine, France) was used for the induction of hyperinsulinemia. At reached steady state, in vivo insulin-stimulated glucose uptake in tissues was determined by a 10 µCi bolus injection of 2-[$^{14}$C] deoxyglucose. After 30 min, mice were rapidly killed by cervical dislocation and tissues removed and stored at −80 °C until use. Glucose concentration was measured using the glucose oxidase method (GLU, Roche Diagnostics, Rotkreuz, Switzerland) and insulin using an ELISA commercial kit (CrystalChem Inc., Downers Grove, IL). Insulin tolerance test (ITT) was performed, as previously described[53]. Briefly, glycemia was monitored in mice after an intraperitoneal bolus of insulin as indicated in the legend of figures representing ITT experiments.

**LPS treatment.** Eight-week-old male mice were treated with saline or LPS (300 µg/kg/day, Lipopolysaccharides from *Escherichia coli* 0127:B8, ref. L3129, Sigma-Aldrich) administered for 28 days using osmotic delivery pumps (model 2004; Alzet) inserted subcutaneously into the backs of animals.

**Kinase activity profiling.** Protein Tyrosine Kinase (PTK) activity profiling was performed using "Pamgene Station". A total of 5 µg of protein extracted from liver (10-week-old $Ptprg^{-/-}$ and $Ptprg^{+/+}$ fed on a standard diet) using RIPA buffer and processed with PamChip4 microarray chips, provided by PamGene (PamGene International, the Netherlands) and printed at their facilities.

**Statistical analysis.** Data sets were analyzed for statistical significance using PRISM (GraphPad, San Diego, CA) for a two-tail unpaired Student's t test when two groups were compared or one-way ANOVA (Tukey's post test) when three or more groups were compared.

**Data availability.** All data generated or analysed in this study are included in this published article (and its supplementary information files). All relevant data are available from the authors.

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

## Acknowledgements

We thank Ariane Widmer, Anne Charollais, and Carolyn Heckenmeyer in the Coppari laboratory for their technical support and Drs. Claes Wollheim, Françoise Rohner-Jeanrenaud, and Michael Cowley for suggestions and critical reading of the manuscript. We thank Dr. Johan Auwerx for providing liver samples from NR-treated mice and their controls. This work was supported in part by Coordenação de Aperfeiçoamento de Pessoal de Nível Superior (CAPES graduate student fellowship to R.M.I.), the Bo & Kerstin Hjelt Foundation (research grant to G.R. and S.L.), and European Commission (Marie Curie Career Integration Grant 320898 and ERC-Consolidator Grant 614847), the Swiss Cancer League (KLS-3794-02-2016-R), the Louis-Jeantet Foundation, the Gertrude von Meissner Foundation, and the Fondation Pour Recherches Medicales of the University of Geneva to R.C., and the Swiss National Science Foundation (310030_169966/1 to R.C. and 314730-166609 to F.N.).

## Author contributions

Conceptualization: R.C., X.B., and G.R.; Investigation: X.B., G.R., C.V.-D., J.A., R.M.I., E. A., S.L., D.K., S.F., S.C., M.T., N.G., S.H., and F.N.; Writing—Original Draft: R.C.; Writing—Review & Editing: R.C., X.B., and G.R.; Funding Acquisition: R.M.I., G.R., R.C., and F.N.; Data Curation: R.C., X.B., F.N., and G.R.; Supervision: R.C. and G.R.

## Additional information

**Competing interests:** The authors declare no competing financial interests.

