## [Peer Review File · Nature Communications]

Reviewers' comments:

Reviewer #1 (expert in liver glucose metabolism)(Remarks to the Author):

Authors tested hepatic role of protein tyrosine phosphatase receptor gamma of obesity-induced type 2 inflammation to type 2 diabetes using Protein Tyrosine Phosphatase Receptor Gamma (PTPRG) global KO mice with series of sophisticated adenoviral overexpression and other experiments affecting insulin and NFkB mediated inflammatory signals on HFD and LPS treatment concluding that PTPRG, especially hepatic PTPRG is a novel and key mediator of inflammation-mediated insulin resistance.

They also showed some supporting association data from human NASH samples and therapeutic effects of SIRT1 on this system.

Basically all the experiments were carefully and logically set up and their conclusions are supported by numerous consistent data in multi-faceted aspects. However, this reviewer raises some fundamental questions and comments as follow.

General comments

1. The differences between PTPRG WT and KO mice except EGP and GIR data from clamp studies are not very strong and required statistical significances. Overall, this reviewer agrees that PTPRG is involved in HFD insulin resistance, but is not convinced of hepatic PTPRG as the sole culprit even after numerous fine experiments until I see the data from liver-specific KO mice and ideally negative data from brain-, adipose- and skeletal muscle- specific KO. This impression is inevitable based upon recent notification of cross talk of organs on this kind of issue. As for contribution of brain, hepatic vagotomy could help. As for liver-specificity, PTPRG KD experiment to show the amelioration should be shown as mirror image of restoration in KO. Authors also need to show some data on discrimination of hepatocytes and non-hepatocytes, especially highlighting roles of Kupffer cells.

2. The unsolved questions about PTPRs are their ligands. They conclude that inflammatory signal turns on hepatic PTPRG and liver-intrinsically causes insulin resistance, which do not shed light on this important question. If authors are working on the important role of PTPRG, some extracellular signaling through this receptor should be involved. To strengthen their conclusions, overexpression of mutant PTPRG lacking extracellular domain, especially CAH domain still holding PTP activity should be tested to restore the phenotype. Authors should at least discuss comparison with other PTP(R)s,

Specific comments

1. The title and the comments P4 I79~ on diabetes are not adequate. Blood sugars are not high enough for diabetes. Glycohemoglobin should be shown. Replacement by "insulin resistance" might be proper.

2. As my impression, the pattern of amelioration of insulin resistance by loss of function PTPRG on ITT experiments implicates peripheral rather than hepatic insulin signaling. It also raises another question about hepatic responsibility. SupFig3b ITT should be shown by %, statistical significance might be gone.

3. Mutant I κ B α experiment should be extended to in vivo using adenoviral overexpression for more convincing data.

4. Promoter analysis should be more extensive and solid.

5. The data from C2C12 as a muscle cell line are fine, but confusing considering their conclusions. Another liver cell line might be used.

6 Is there any supporting human data on SNP of PTPRG?

Reviewer #2 (expert in gluconeogenesis) (Remarks to the Author):

This is a potentially interesting study showing that hepatic protein tyrosine phosphatase receptor- γ (PTPR- γ), induced by NF- κ B activation, deteriorates whole body insulin sensitivity and glucose tolerance by impairing hepatic insulin signaling. The data are convincing, but not currently sufficient to fully support the authors' proposal. I describe my specific concerns below.

My two major concerns are as follows: 1) Does PTPR- γ negatively regulate insulin sensitivity and glucose tolerance constitutively, and not in an obesity-induced inflammation-dependent manner? 2) It is not clear how PTPR- γ deteriorates insulin signaling. Is its phosphatase activity required? If so, what is the substrate of PTPR- γ ? Is the substrate the insulin receptor, IRS-1/2, and/or others?

Major points

1. PTPR- γ knockout mice exhibited lower blood glucose and plasma insulin levels compared to control mice under both standard chow- and high-caloric diet-fed conditions. These data seem to indicate that PTPR- γ negatively regulates insulin sensitivity and glucose tolerance constitutively, not in an obesity-induced inflammation-dependent manner. To clarify this issue, the metabolic phenotype of PTPR- γ knockouts as well as insulin signaling in the liver, skeletal muscles, and adipose tissues should be analyzed in both STD- and HCD-fed mice (Figure 2c–g, Supplementary Figure 3a–e).
2. PTPR- γ heterozygous knockout mice fed STD or HCD may exhibit dose-dependent effects in their metabolic phenotype as well as hepatic insulin signaling. Such data should be provided to strongly support the authors' proposal.
3. The molecular mechanism by which PTPR- γ deteriorates insulin sensitivity was not clarified. To address this issue, whether the phosphatase activity of PTPR- γ is necessary to induce metabolic derangement in vivo and insulin signaling defects in vitro (in primary hepatocytes) is evaluated based on the expression of phosphatase-deficient mutants or wild-type PTPR- γ in PTPR- γ -deficient liver or primary hepatocytes. If phosphatase activity is necessary, PTPR- γ 's substrate should be determined in vitro and/or in vivo. At minimum, the insulin receptor (IR) and IRS-1/2 should be evaluated as substrate candidates.
4. In Figure 3j, overexpression of PTPR- γ in the liver of FVB mice reduced glucose disposal during insulin clamps, suggesting impaired insulin-dependent glucose uptake in the skeletal muscle. The authors should examine muscle insulin signaling and explain why hepatic deletion of PTPR- γ affects muscle glucose disposal.
5. SIRT1-dependent regulation of PTPR- γ expression and insulin sensitivity is of interest. However, in vivo evidence suggesting such regulation is not provided. Does resveratrol administration in mice on STD or HCD reduce hepatic Ptprg expression? Does resveratrol administration fail to reverse the metabolic derangement induced by hepatic overexpression of PTPR- γ ? To strengthen the proposal, these issues should be addressed.
6. In Figure 4k, the two graphs should be combined to easily compare the data from the 4 groups. Although Ptprg $^{-/-}$ mice fed an STD exhibited improved glucose metabolism and hepatic insulin signaling, is the insulin sensitivity in knockouts similar to that of Ptprg $^{+/+}$ mice?

Minor points

1. In Figure 1a and b, the differences in PTPR- γ mRNA levels were only ~2-fold. Protein abundances should be provided.
2. The correlative evidence between PTPRG mRNA abundance, NAFL, and NASH in humans is interesting. In Figure 1c, in addition to NASH, PTPRG mRNA abundance in NAFL subjects should be provided.
3. The authors mentioned that the mRNA abundances of IkBA and MnSOD are correlated with inflammation. Related references should be provided.
4. In Supplementary Figure 2e–g and i, data from STD-fed mice should be provided.
5. In Supplementary Figure 2j, to directly compare the mRNA levels of these two groups, the two

individual graphs should be combined.

6. To evaluate tyrosine phosphorylation of insulin receptor in Figure 2g and Supplementary Figure 3f and 4b, the authors used phospho-IR^{Tyr1361} antibody. This antibody can detect insulin-induced phosphorylation of IR in both mouse liver and primary cultured hepatocytes, but the signal intensity appears to be weak. Phosphorylation of this site may not necessarily reflect overall phosphorylation. Overall tyrosine phosphorylation of IR should be evaluated using anti-IR immunoprecipitates and anti-phosphotyrosine antibodies.

7. In the analysis of insulin signaling (Figure 2g and Supplementary Figure 3f and 4b), tyrosine phosphorylation and protein levels of IRS-1/2, as well as the association of the p85 subunit of PI 3-kinase with IRS-1/2, should be evaluated.

8. In Figure 4a, a description of IkBA DN should be provided.

9. In Figure 4a and b, IkBA DN may affect Ptpg mRNA expression in control cells. To address this issue, the two graphs should be combined.

10. In Figure 4d, NF- κ B occupancy on the Ptpg promoter in the liver of HCD-fed mice should be evaluated.

Reviewer #3 (expert in inflammation and diabetes)(Remarks to the Author):

In this study, the authors show that inflammation in obese/T2D in mice induces Protein Tyrosine Phosphate Receptor Gamma (PTPR- γ), and, that this requires the binding of NF- κ B to the PTPR- γ promoter. Global knockout of PTPR- γ reduces glycemia and insulin levels and protects from the development of T2D, primarily via the reduction of endogenous glucose generation. Importantly, overexpression of PTPR- γ in the liver in global PTPR- γ deficient mice to levels observed in obesity leads to systemic and hepatic insulin resistance, indicating a primary role for hepatic PTPR- γ in the link between obesity-induced inflammation and T2D. Data showing that PTPR- γ positively correlates with indices of inflammation and insulin resistance in humans suggest that this pathway maybe also operational in human obesity/insulin resistance/T2D. This is a well conducted and elegant study, and, the experiments are carefully chosen to arrive at the said conclusions. The mice experiments provide proof of concept while the data on humans points to the relevance of the pathway in human obesity/T2D and suggests the potential of hepatic PTPR- γ as a new therapeutic target to treat T2D. There are however, certain issues that needs to be addressed/clarified.

Only a limited number of inflammatory cytokines are studied in both mice and humans. The authors need to expand to include a larger panel of the usual suspects of inflammatory cytokine in the analysis.

The authors indicate that binding of NF- κ B to the PTPR- γ promoter is required for PTPR- γ expression and subsequent inflammation. However, it is known that NF- κ B directly induces inflammation via binding to the promoter region of many inflammatory cytokines (e.g. TNF- α , IL-6 etc), and, inflammation activates NF- κ B. The authors need to clarify and discuss this relationship in the context of their data.

Was the improvement in glucose and insulin tolerance observed only in HCD fed Ptpg^{-/-} mice? (Supplementary figures 3a, 3b), given that other metabolic parameters were improved in Ptpg^{-/-} mice fed a NCD (Figures 2a, 2b)?

There is really no change in the insulin-induced glucose disposal in the Ptpg^{-/-} mice (Fig. 2d). The authors state that it is slightly increased; this is not convincing.

Not convinced that you can unequivocally state that "hepatic PTPR- γ underlies the glucose and insulin phenotypes caused by whole-body PTPR- γ deletion. Liver maybe primary, but contributions from adipose and muscle cannot be ignored. There is evidence for this in the 2-DG uptake studies

and the increased expression in adipose and muscle under HCD. Insulin enhances glucose uptake in the adipose and muscle and suppresses endogenous glucose production from the liver suggesting that all three metabolically active tissues are involved in the link between PTPR- γ /inflammation/T2D. The authors have not comprehensively followed through in these tissues, but focuses mainly in the liver.

Figure 3F: hyperglycemia is not convincing in the Ad- Ptprg mice. The authors may want to include the GTT for these mice.

In Supplementary Fig 5d and Fig. 4a, 4b, why wasn't an adipocyte cell line included in these studies? What was the rationale for treating with TNF- α and LPS for 1 day and with IL-1 for 1 hour? Long time frame of 1 day may not necessarily indicate direct effects, could be a secondary effect?

Page 6, line 140 (Fig. 3j,k j) should be (Fig. 3j, k)?

The methods for Animal studies state that only littermates of the same sex were randomly assigned to experimental groups and compared to each other. However, the figure legends do not indicate which sex was used in the various experiments. Were there gender differences in the expression of PTPR- γ in obese mice and in humans?

The figure legends do not indicate where appropriate the number of times the experiment was repeated.

Not all figures legends indicate the number of replicates used, i.e. N=? for all panels in the figure.

Need to be consistent when expressing relative mRNA values; either specific mRNA level/18s or mRNA (arbitrary units). Authors use both in various experiments.

What was the standard diet used? kcal? Were the diets matched for kcal and only different in the fat content?

Need to be consistent with the abbreviation PTPR- γ versus Ptprg.

Reviewer #4 (expert in inflammation and obesity) (Remarks to the Author):

In this study, Brenachot et al., hepatic PTPRg is upregulated in obesity and by inflammatory signals and its deletion provides protection against obesity-induced as well as LPS-driven inflammation and insulin resistance. The authors also show that PTPRg expression is upregulated in humans with NASH and correlates positively with both inflammatory indicators such as I κ B α and insulin resistance. While the studies were conducted in a whole body deletion model, the authors did virally re-express PTPRg in the liver and demonstrated that this intervention was sufficient to revert the impact of whole body deficiency on the primary metabolic phenotype. Mechanistically, the authors addressed two aspects. First they provide evidence, in cellular systems, that PTPRg regulation by inflammatory mediators is dependent on NF- κ B signaling. Second, they provided evidence that suppression of SIRT1 results in increased PTPRg expression and resveratrol treatment attenuates LPS-induced PTPRg and inflammatory marker expression. Finally, they treated obese mice with nicotinamide riboside, which also resulted in reduced PTPRg and inflammatory marker expression in the liver. As a result, the authors suggest that PTPRg can be a potential therapeutic target to control obesity-induced inflammation and metabolic pathologies. These are interesting studies and contribute to the understanding of immunometabolic pathologies underlying obesity and diabetes. The authors may consider the following points to strengthen their

conclusions and overall impact of the manuscript.

PTPRg expression is also increased in adipose and muscle tissue. Do the authors suggest that it has no function at these sites? Have they attempted to examine inflammatory/metabolic status or insulin action at these sites with any additional experiments? The clamp study shows a major improvement in glucose uptake in the adipose tissue. Can the authors quantitatively compare the glucose uptake differences when they re-expressed PTPRg in the liver? Have the authors examined the inflammatory status in the adipose tissue? These questions are raised to stimulate the authors to consider the possibility that while important, perhaps even major, there may be other sites, which also contribute to the deficiency phenotype. In particular, adipose tissue (and adipocytes) may deserve the attention of the investigators.

While the NF- κ B mediated regulation is sufficiently demonstrated in cells, the same is not quite the case for SIRT1. Also, whether these pathways do act in vivo in a manner similar to cells is examined in a rather limited and indirect manner. It is difficult to make strong conclusions based on the use of resveratrol or nicotinamide riboside. It would be desirable to build stronger links between observation in cells and that of tissues in vivo. Alternatively, the author may want to generate additional insight about the mechanism by which PTPRg impairs metabolic function in vivo. Addressing one of these aspects would have a strong impact on the conclusions of the paper.

Minor points:

The authors may want to use a more broad and balanced presentation of the field, especially in their introduction and abstract and add some more to their discussion to position their findings and address some of the caveats. For example:

Line 26: "Yet, the inflammatory effector(s) linking obesity to insulin resistance is unknown"

This statement is way too strong. Many things are known so stating that it is unknown is not appropriate. Also, it is highly unlikely that there will be one molecular mediator, including PTPRg, to justify the use of "the mediator" terminology.

Line 44-46: "However, pharmacological approaches aimed at targeting the aforementioned mechanisms have had inadequate outcomes or still need to be translated into the clinic 8-14"

Here the authors should represent the literature in a more balanced manner. First, there is now plenty of evidence coming from the humans. There are recent and comprehensive reviews, for example one just published in Nature (PMID: 28179656), to find all relevant references, including those with IL1B blockade, TNF antagonists, CD44, lipid mediators, resolvins, atypical erythropoietin receptor targeting peptide, and others. Second, there are emerging and new strategies to combat inflammation in obesity which has provided exciting benefits against insulin resistance and diabetes in proof-of-principle studies, for example amlexanox from Saltiel's group (PMID:28683283) and immune cell-mediated strategies (PMID:28685960). I would suggest this sentence to be revised into something like "While pharmacological approaches aimed at targeting the aforementioned mechanisms have had encouraging outcomes there is still an unmet need for new, broad and effective tools for translation into the clinic".

I would also advise revising the reference list used here. For example, refs 11-14 are all on salsalate. Why not to use one primary paper and perhaps one review and expand the other exciting targets that are being used or tested in humans in this reference list.

The authors also have presented the NF κ B pathway as if it is the only signaling pathway relevant to obesity-induced inflammation. While this is a very critical mechanism, there are several other

key mediators, such as JNK and p38 as well as inflammasome which are not measured or mentioned.

In general a more balanced tone and citations and some attention to the shortcomings of the current study would be a prudent and would not diminish the value of the current study.

Point-by-point response to reviewers' criticisms:

Reviewer #1

1) “The differences between PTPRG WT and KO mice except EGP and GIR data from clamp studies are not very strong and required statistical significances. Overall, this reviewer agrees that PTPRG is involved in HFD insulin resistance, but is not convinced of hepatic PTPRG as the sole culprit even after numerous fine experiments until I see the data from liver-specific KO mice and ideally negative data from brain-, adipose- and skeletal muscle- specific KO. This impression is inevitable based upon recent notification of cross talk of organs on this kind of issue. As for contribution of brain, hepatic vagotomy could help. As for liver-specificity, PTPRG KD experiment to show the amelioration should be shown as mirror image of restoration in KO. Authors also need to show some data on discrimination of hepatocytes and non-hepatocytes, especially highlighting roles of Kupffer cells.”

In the *in vivo* context, the liver of mice lacking *Ptprg* displays increased insulin sensitivity (Fig. 2h, Supplementary Fig. 3h and Supplementary Fig. 4a) while the liver of mice overexpressing hepatic *Ptprg* is insulin resistant (Supplementary Fig. 5d). In the *ex-vivo* context, primary hepatocytes lacking *Ptprg* have increased insulin signaling (Supplementary Fig. 4b). These data would suggest that the negative effect of PTPR- γ on hepatic insulin action is cell-autonomous.

Nevertheless, we totally agree with this Reviewer that to directly test the role of PTPR- γ in specific tissues/cells on glucose and insulin homeostasis studies in mice lacking *Ptprg* in a tissue/cell-specific fashion (e.g. adipocytes, neurons, skeletal muscle, immune/Kupffer cells) are required. Unfortunately, we cannot perform these experiments because Cre-conditional *Ptprg* null mice are currently unavailable.

To circumvent this technical limitation, we performed additional glucose tracing analyses and biochemical assays to carefully assess glucose uptake and insulin signaling in extra-hepatic tissues.

Interestingly, the increased glucose disposal observed in adipose tissue and skeletal muscle of mice lacking PTPR γ during the clamp appears to be independent to changes in insulin action at these sites. In fact, increased glucose disposal in fat and skeletal muscle of mice lacking PTPR γ was also observed in the basal state (Fig 2e-f). Furthermore, insulin-induced phosphorylation of AKT and IR in these tissues was not different between *Ptprg*^{-/-} and *Ptprg*^{+/+} mice (Supplementary Fig.3f,g).

We also measured glucose disposal in tissues of mice overexpressing PTPR γ restrictedly in the liver. Almost as a mirror image of the PTPR γ null mice data, hepatic PTPR γ overexpression caused a reduction of glucose disposal in adipose tissue (but not in skeletal muscle) (Supplementary Fig. 5b). Insulin-induced phosphorylation of AKT

and IR in adipose tissues was not different between mice overexpressing hepatic PTPR γ and their controls (Supplementary Fig. 5c).

Our new data shown in Supplementary Fig.3f,g clearly indicate that insulin action is not enhanced in these extra-hepatic tissues. Even though we show that i) re-expression of *Ptprg* in liver of mice that otherwise lack *Ptprg* globally is sufficient to correct glucose/insulin phenotype caused by *Ptprg* deficiency (Fig. 3a-d) and that ii) overexpression of *Ptprg* in liver of wild-type mice is sufficient to deteriorate insulin sensitivity (Fig 3i-k and Supplementary Fig.5d) we cannot exclude that *Ptprg* in other tissues might also affect glucose metabolism and insulin sensitivity. In the revised manuscript we have amended the Discussion section by discussing the limitation of our findings and the need for future studies in mice lacking *Ptprg* in a tissue/cell-specific fashion.

2) “The unsolved questions about PTPRs are their ligands. They conclude that inflammatory signal turns on hepatic PTPRG and liver-intrinsically causes insulin resistance, which do not shed light on this important question. If authors are working on the important role of PTPRG, some extracellular signaling through this receptor should be involved. To strengthen their conclusions, overexpression of mutant PTPRG lacking extracellular domain, especially CAH domain still holding PTP activity should be tested to restore the phenotype. Authors should at least discuss comparison with other PTP(R)s.”

We agree with this Reviewer that our results will spur interests in identifying endogenous PTPR γ ligand(s). Interestingly, several members of the contactin family have already been suggested as endogenous PTPR- γ ligands (Kianfar et al., 2017; Nikolaienko et al., 2016).

To test whether these contactins i) cause inactivation of PTPR- γ , ii) affect insulin and glucose homeostasis *in vivo*, and iii) whether their effect requires PTPR γ will necessitate extensive studies in several genetically-engineered animal models overexpressing each one of these contactin proteins and bearing or lacking PTPR γ . These experiments will require several years before completion. Thus, we think that they are beyond the scope of the current study.

Nevertheless, in the Discussion section we introduced the possibility that these proteins could inhibit PTPR γ and improve T2DM. Also, we compared our PTPR γ results with data gathered from mice lacking PTP1B.

3) “The title and the comments P4 179~ on diabetes are not adequate. Blood sugars are not high enough for diabetes. Glycohemoglobin should be shown. Replacement by "insulin resistance" might be proper.”

We followed this Reviewer’s suggestion and changed the title and narrative accordingly.

4) “As my impression, the pattern of amelioration of insulin resistance by loss of function PTPRG on ITT experiments implicates peripheral rather than hepatic

insulin signaling. It also raises another question about hepatic responsibility. SupFig3b ITT should be shown by %, statistical significance might be gone.
”

Although indicative, results from ITT experiments are not sufficient for drawing reliable conclusions on whole-body insulin sensitivity. Not to mention that results from these tests cannot be used for drawing any conclusion on insulin sensitivity at the single tissue level.

The state-of-the-art approaches for assessment of insulin sensitivity at the organismal and at the single tissue level include hyperinsulinemic/euglycemic clamp and biochemical studies. We performed these state-of-the-art experiments in loss- and gain-of-function contexts. Our new and previous data provide with several independent evidences that PTPR γ negatively regulates insulin sensitivity in hepatocytes (Fig 2g,h and Supplementary Fig.4a,b). Also our new and previous data indicate that PTPR γ does not regulate insulin sensitivity in adipose tissue and skeletal muscle (Fig. 2e,f and Supplementary 3f,g).

5) “Mutant I κ B α experiment should be extended to in vivo using adenoviral overexpression for more convincing data.”

We were unable to perform these experiments and in a new paragraph of the revised manuscript we discussed the limitation of our study.

6) “Promoter analysis should be more extensive and solid.”

To consolidate our result of NF- κ B occupancy on *Ptprg* promoter, we extended our analysis *in vivo* by performing chromatin immunoprecipitation (ChIP) assays on liver of HCD-fed mice. Our new data indicate that NF- κ B occupancy in the promoter region of *Ptprg* in mouse liver is increased in the context of HCD-feeding (Fig 4e) and in the context of LPS treatment (Fig. 4d). These new and the numerous previous data further support our overall conclusion that *Ptprg* is transcriptionally regulated by NF- κ B.

7) “The data from C2C12 as a muscle cell line are fine, but confusing considering their conclusions. Another liver cell line might be used.”

We have several *in vitro* and *in vivo* evidences supporting the conclusion that inflammation induces PTPR γ expression in hepatocytes. First, our human data indicate that *PTPRG* expression correlates with inflammatory markers (Fig. 1d,e and Supplementary Fig. 1a-e). Second, in mice hepatic PTPR γ expression is enhanced in the context of increased inflammation (Fig. 1 a,b). Third, in mice NF- κ B occupancy of PTPR γ promoter is enhanced in liver in the context of increased inflammation (LPS treatment and HCD feeding) (Fig. 4d,e). Fourth, several inflammatory cues induce PTPR γ expression in hepatocytes (Fig. 4a) and this effect requires NF- κ B (Fig. 4a). Hence, we believe that results from experiments performed in another hepatic cell line are likely to yield incremental insights.

As for the use of a muscle cell line: we observed changes in glucose uptake in skeletal muscle of mice lacking PTPR γ . Also, PTPR γ expression is induced by HCD feeding and LPS treatment. Due to the lack of mice lacking PTPR γ in this tissue, the relevance of PTPR γ in skeletal muscle is currently unknown. In this study we provided evidences that the regulation of PTPR γ expression in skeletal muscle cells is similarly controlled by NF-kB (Fig 4b). These results will spur interests in generating and studying mice lacking PTPR γ in skeletal muscle also because lack PTPR γ enhances glucose uptake in this tissue in an insulin-independent fashion (Fig. 2e,f and Supplementary 3g).

8) “Is there any supporting human data on SNP of PTPRG?”

To the best of our knowledge we are not aware of PTPRG SNPs and their association with insulin sensitivity. However, we have supporting human data shown in Fig. 1 and Supplementary Fig. 1.

Reviewer #2

Major Points:

1) “Does PTPR- γ negatively regulate insulin sensitivity and glucose tolerance constitutively, and not in an obesity-induced inflammation-dependent manner?”

Combined with our previous data, our new results demonstrate that PTPR- γ is a negative regulator of hepatic insulin signaling in both physiological and obesity/inflammation contexts (Fig. 2c,g,h; Fig. 3g-k; Supplementary Fig.4a,b; Supplementary Fig.5d).

2) “It is not clear how PTPR- γ deteriorates insulin signaling. Is its phosphatase activity required? If so, what is the substrate of PTPR- γ ? Is the substrate the insulin receptor, IRS-1/2, and/or others?”

Enzymatic assays indicate that PTPR- γ exerts phosphatase activity against several targets with an apparent selective preference for peptides with N-terminal acidic and C-terminal basic sequences relative to phosphorylated tyrosine (Barr et al., 2009). Among these targets is the insulin receptor (IR) (Barr et al., 2009) and Janus Kinase 2 (JAK2) (Mirenda et al., 2015). Our new and previous data are in line with these results as we observed changes in phosphorylation status of IR in liver of mice lacking or overexpressing PTPR- γ . Please, see our new data in Supplementary Fig 4b and our previous data in Fig. 2h.

In addition, we performed an unbiased assay to measure the activity of phosphatases/kinases in lysates obtained from liver samples of mice lacking PTPR- γ and their controls. By using arrays with immobilized tyrosine peptide substrates we were able to identify known (e.g. JAK2) and potentially new targets of PTPR- γ . Our new results shown in Fig. 2i and Supplementary Table 1 combined with our previous data shown in Fig. 2g,h indicate that PTPR- γ affects glucose metabolism by altering phosphorylation status of IR, PI3K, AKT and also other proteins such as LAT, PGFRG, EPHB1, ANXA1,

FGFR3, JAK2, FES, VGFR2, LCK, and JAK1. To determine the role of specific PTPR- γ domains and the relevance of each one of these proteins in mediating the effects of PTPR- γ on glucose metabolism *in vivo* extensive efforts are required. Thus, we believe that our new results will pave the way for several future studies.

3) “PTPR- γ knockout mice exhibited lower blood glucose and plasma insulin levels compared to control mice under both standard chow- and high-caloric diet-fed conditions. These data seem to indicate that PTPR- γ negatively regulates insulin sensitivity and glucose tolerance constitutively, not in an obesity-induced inflammation-dependent manner. To clarify this issue, the metabolic phenotype of PTPR- γ knockouts as well as insulin signaling in the liver, skeletal muscles, and adipose tissues should be analyzed in both STD- and HCD-fed mice (Figure 2c–g, Supplementary Figure 3a–e).”

Please, see answer to your comment #1 mentioned above.

4) “PTPR- γ heterozygous knockout mice fed STD or HCD may exhibit dose-dependent effects in their metabolic phenotype as well as hepatic insulin signaling. Such data should be provided to strongly support the authors’ proposal.”

We thank this Reviewer for this suggestion. We also wanted to address this same question and started our studies in heterozygous knockout mice. However, we realized that these mice have normal *Ptprg* expression in liver, muscle, and adipose tissue. Please, see these data below.

Figure legend: *Ptprg* mRNA content in tissues of 10-week-old wild-type male mice (*Ptprg*^{+/+}; n=5) and their male littermates heterozygous for the *Ptprg* null allele (*Ptprg*^{+/-}; n=5) fed on a standard diet.

Therefore, these heterozygous mice cannot be used for determining whether the effect of PTPR- γ in glucose/insulin balance is dose-dependent.

5) “The molecular mechanism by which PTPR- γ deteriorates insulin sensitivity was not clarified. To address this issue, whether the phosphatase activity of PTPR- γ is necessary to induce metabolic derangement *in vivo* and insulin signaling defects *in vitro* (in primary hepatocytes) is evaluated based on the expression of phosphatase-deficient mutants or wild-type PTPR- γ in PTPR- γ -deficient liver or primary hepatocytes. If phosphatase activity is necessary, PTPR- γ 's substrate should be determined *in vitro* and/or *in vivo*. At minimum, the insulin receptor (IR) and IRS-1/2 should be evaluated as substrate candidates.”

Please, see answer to your comment #2.

6) “In Figure 3j, overexpression of PTPR- γ in the liver of FVB mice reduced glucose disposal during insulin clamps, suggesting impaired insulin-dependent glucose uptake in the skeletal muscle. The authors should examine muscle insulin signaling and explain why hepatic deletion of PTPR- γ affects muscle glucose disposal.”

To address this important issue, we added new data in Supplementary Fig. 5b. Our new results indicate that hepatic overexpression of PTPR- γ does not affect glucose uptake in skeletal muscle. However, it dampens glucose uptake in adipose tissue (Supplementary Fig 5b). Interestingly, hepatic overexpression of PTPR- γ does not affect insulin sensitivity in this tissue (Supplementary Fig. 5c). The mechanisms by which hepatic PTPR- γ regulates glucose metabolism at distant sites (e.g. adipose tissues) are currently being investigated. For example, we measured the level of circulating TNF- α and interleukin-6 (IL-6) in the loss- and gain-of-function contexts and found the levels of these potential mediators unchanged (Supplementary Fig. 6a,c,d). Thus, the mechanisms by which hepatic PTPR- γ regulates glucose metabolism at distant sites are yet to be identified.

7) “SIRT1-dependent regulation of PTPR- γ expression and insulin sensitivity is of interest. However, *in vivo* evidence suggesting such regulation is not provided. Does resveratrol administration in mice on STD or HCD reduce hepatic Ptprg expression? Does resveratrol administration fail to reverse the metabolic derangement induced by hepatic overexpression of PTPR- γ ? To strengthen the proposal, these issues should be addressed.”

We have two set of data indicating that SIRT1 regulates PTPR- γ expression also *in vivo*. First, deletion of SIRT1 in specific subsets of neurons within the hypothalamus leads to increased PTPR- γ expression. Please, see these new data shown in Supplementary Fig. 6i. Second, mice treated with the SIRT1 activator NR have increased PTPR- γ expression. Please, see these data shown in Supplementary Fig. 6j.

8) In Figure 4k, the two graphs should be combined to easily compare the data from the 4 groups. Although *Ptprg*^{-/-} mice fed an STD exhibited improved glucose metabolism and hepatic insulin signaling, is the insulin sensitivity in knockouts similar to that of *Ptprg*^{+/+} mice?

We removed the ITT data because we have assessed insulin action by the state-of-the-art approaches. Please, see answer to Reviewer #1 comment #4.

Minor Points:

1) “In Figure 1a and b, the differences in PTPR- γ mRNA levels were only ~2-fold. Protein abundances should be provided.”

We have tested several commercially available antibodies. We have also attempted to generate our own antibodies. Unfortunately, all of these reagents gave rise to either no signal or positive signal(s) when used in samples from wild-type mice. However, these signals were not specific to PTPR- γ because were also visible when these antibodies were used in samples obtained from *Ptprg*^{-/-} mice. Hence, we concluded that all of these reagents are unsuitable for visualizing and quantifying PTPR- γ protein content.

In the revised narrative we mentioned the following: “We were unable to assess PTPR- γ protein content due to the lack of specific PTPR- γ antibodies (data not shown).”

2) “The correlative evidence between PTPRG mRNA abundance, NAFL, and NASH in humans is interesting. In Figure 1c, in addition to NASH, PTPRG mRNA abundance in NAFL subjects should be provided.”

We followed this Reviewer’s suggestion. Hence, in the revised manuscript PTPRG mRNA contents in healthy, NAFL, and NASH subjects are shown in Fig. 1c. Our data indicate that PTPRG mRNA contents increase proportionally to disease severity. Indeed, NASH subjects display the highest while NAFL subjects an intermediate level of increased PTPRG mRNA contents compared to healthy individuals.

3) “The authors mentioned that the mRNA abundances of IkBA and MnSOD are correlated with inflammation. Related references should be provided.”

These are established NF- κ B targets. We changed the narrative and added the reference accordingly.

4) “In Supplementary Figure 2e–g and i, data from STD-fed mice should be provided.”

We thank this Reviewer for these suggestions.

Pertinent to his/her suggestion to perform indirect calorimetry analyses in STD-fed mice lacking PTPR- γ and their controls we would like to point out the following. STD-fed mice lacking PTPR- γ have normal body weight, body composition, and energy

intake (Supplementary Fig. 2a, b, d). Thus, based on the first law of thermodynamics their energy output must be normal. In fact, indirect calorimetry data from HCD-fed mice confirm this notion because mice lacking PTPR- γ have body weight, body composition, and energy intake similar to their wild-type controls and their energy output is also similar to controls (Supplementary Fig. 2a, b, d, e, f, g).

Pertinent to his/her suggestion to perform UCP1 immunohistochemistry in samples of STD-fed mice lacking PTPR- γ and their controls, please see these new data shown in Supplementary Fig. 2i.

5) “In Supplementary Figure 2j, to directly compare the mRNA levels of these two groups, the two individual graphs should be combined.”

We followed this Reviewer’s suggestion. Please, see revised Supplementary Fig. 2j.

6) “To evaluate tyrosine phosphorylation of insulin receptor in Figure 2g and Supplementary Figure 3f and 4b, the authors used phosho-IRTyr1361 antibody. This antibody can detect insulin-induced phosphorylation of IR in both mouse liver and primary cultured hepatocytes, but the signal intensity appears to be weak. Phosphorylation of this site may not necessarily reflect overall phosphorylation. Overall tyrosine phosphorylation of IR should be evaluated using anti-IR immunoprecipitates and anti-phosphotyrosine antibodies.”

We performed these experiments. Please, see new data shown in Supplementary Fig. 3h.

7) “In the analysis of insulin signaling (Figure 2g and Supplementary Figure 3f and 4b), tyrosine phosphorylation and protein levels of IRS-1/2, as well as the association of the p85 subunit of PI 3-kinase with IRS-1/2, should be evaluated.”

To gain more mechanistic insights into the function of PTPR- γ , we measured the activity of phosphatases/kinases in lysates obtained from liver samples of mice lacking PTPR- γ and their controls. By using arrays with immobilized tyrosine peptide substrates we were able to identify known (e.g. JAK2) and putative new targets of PTPR- γ . Our new results shown in Fig. 2i and Supplementary Table 1 combined with our previous data shown in Fig. 2g,h and Supplementary Fig. 3j indicate that PTPR- γ affects hepatic insulin signaling and glucose metabolism by altering phosphorylation status of proteins whose metabolic role is known (e.g. IR, P85A, AKT) and also of others whose function on hepatic insulin signaling and glucose metabolism is inadequately understood (e.g. LAT, PGFRG, EPHB1, ANXA1, FGFR3, JAK2, FES, VGFR2, LCK, and JAK1). Together, our data demonstrate that lack of PTPR- γ greatly improves glucose homeostasis at least in part by enhancing hepatic insulin sensitivity.

8) “In Figure 4a, a description of IkBA DN should be provided.”

To improve clarity, in the text of revised manuscript we included an annotation for the mutant super-repressor allele of I κ B α and its reference. Furthermore, we added a description of I κ B α DN sequence in the figure legend.

9) “In Figure 4a and b, I κ B α DN may affect P tprg mRNA expression in control cells. To address this issue, the two graphs should be combined.”

The graphs have been combined and are shown in Fig. 4a, b.

10) “In Figure 4d, NF- κ B occupancy on the P tprg promoter in the liver of HCD-fed mice should be evaluated.”

We have done this experiment. Please, see new data in Fig. 4e.

Reviewer #3

1) “Only a limited number of inflammatory cytokines are studied in both mice and humans. The authors need to expand to include a larger panel of the usual suspects of inflammatory cytokine in the analysis.”

We thank this Reviewer for this suggestion. To address this issue, we added new human data in Figures 1e, and Supplementary Fig. 1a, b and also new murine data in Supplementary 6c,d,e.

2) “The authors indicate that binding of NF- κ B to the PTPR- γ promoter is required for PTPR- γ expression and subsequent inflammation. However, it is known that NF- κ B directly induces inflammation via binding to the promoter region of many inflammatory cytokines (e.g. TNF- α , IL-6 etc), and, inflammation activates NF- κ B. The authors need to clarify and discuss this relationship in the context of their data.”

We thank this reviewer for this comment but we do not see changes in pro-inflammatory inputs in the loss- and gain-of-function models. Please, see data shown in Supplementary 6a-e.

3) “Was the improvement in glucose and insulin tolerance observed only in HCD fed P tprg -/- mice? (Supplementary figures 3a, 3b), given that other metabolic parameters were improved in P tprg -/- mice fed a NCD (Figures 2a, 2b)?”

Please, see answer to comment #1 of Reviewer #2.

4) “There is really no change in the insulin-induced glucose disposal in the P tprg -/- mice (Fig. 2d). The authors state that it is slightly increased; this is not convincing.”

We totally agree with this Reviewer and amended the narrative accordingly.

5) “Not convinced that you can unequivocally state that “hepatic PTPR- γ underlies the glucose and insulin phenotypes caused by whole-body PTPR- γ deletion. Liver maybe primary, but contributions from adipose and muscle cannot be ignored. There is evidence for this in the 2-DG uptake studies and the increased expression in adipose and muscle under HCD. Insulin enhances glucose uptake in the adipose and muscle and suppresses endogenous glucose production from the liver suggesting that all three metabolically active tissues are involved in the link between PTPR- γ /inflammation/T2D. The authors have not comprehensively followed through in these tissues, but focuses mainly in the liver.”

We agree with this comment. To address this issue we carefully assessed insulin signaling in adipose tissue and skeletal muscle of mice lacking *Ptprg*, mice overexpressing *Ptprg* in liver and their controls. Please, see answer to Reviewer #1 comment #1.

6) “Figure 3F: hyperglycemia is not convincing in the Ad- *Ptprg* mice. The authors may want to include the GTT for these mice.”

We performed ITT in these mice (Fig. 3h). Although indicative, results from ITT and GTT experiments are not sufficient for drawing reliable conclusions on whole-body insulin sensitivity. Not to mention that results from these tests cannot be used for drawing any conclusion on insulin sensitivity at the single tissue level. The state-of-the-art approaches for assessment of insulin sensitivity at the organismal and at the single tissue level include hyperinsulinemic/euglycemic clamp and biochemical studies. We performed these state-of-the-art experiments in the gain-of-function contexts (Fig. 3f,g,i,j,k and Supplementary Fig. 5b,c,d).

7) “In Supplementary Fig 5d and Fig. 4a, 4b, why wasn’t an adipocyte cell line included in these studies? What was the rationale for treating with TNF- α and LPS for 1 day and with IL-1 for 1 hour? Long time frame of 1 day may not necessarily indicate direct effects, could be a secondary effect?”

We appreciate these suggestions. As for the adipocyte cell line: we did not want to exclude adipocytes. In fact, we think that performing similar experiments in other cell types, including adipocytes, is very interesting. However, based on the new data shown in Supplementary Fig.3f and Supplementary Fig. 5c indicating that insulin action is not changed in adipose tissue of mice lacking PTPR- γ and also in mice overexpressing PTPR- γ in the liver we decided not to pursue these study here.

8) “Page 6, line 140 (Fig. 3j,k j) should be (Fig. 3j, k)?”

We appreciate this suggestion and made the correction as indicated.

9) “The methods for Animal studies state that only littermates of the same sex were randomly assigned to experimental groups and compared to each other. However, the figure legends do not indicate which sex was used in the various experiments. Were there gender differences in the expression of PTPR- γ in obese mice and in humans?”

We thank this Reviewer for this suggestion. We studied males. We have not studied females and therefore do not know if there are gender differences. These specifications are now included in the Methods section.

10) “The figure legends do not indicate where appropriate the number of times the experiment was repeated.”

To address this issue, we made changes to the figure legends and where appropriate indicated the number of times each experiment was repeated.

11) “Not all figures legends indicate the number of replicates used, i.e. N=? for all panels in the figure.”

To address this issue, we made changes to the figure legends and indicated the number of animals included in each of the experiments.

12) “Need to be consistent when expressing relative mRNA values; either specific mRNA level/18s or mRNA (arbitrary units). Authors use both in various experiments.”

We followed this suggestion and indicated data as relative values.

13) “What was the standard diet used? kcal? Were the diets matched for kcal and only different in the fat content?”

We used standard chow rodent diet RM1 from SDS, Essex, UK. Its Kcal content is 4.198 Kcal/g. We used high-fat diet D12331 from Research Diets, New Brunswick, NJ, USA. Its Kcal content is 5.56 Kcal/g. These specifications are now included in the Methods section.

14) “Need to be consistent with the abbreviation PTPR- γ versus *Ptprg*.”

We used *Ptprg* and PTPR- γ to refer to its gene/mRNA or protein level/function, respectively. We believe this is the right nomenclature for genes or proteins.

Reviewer #4

Major points:

1) “PTPRg expression is also increased in adipose and muscle tissue. Do the authors suggest that it has no function at these sites? Have they attempted to examine inflammatory/metabolic status or insulin action at these sites with any additional experiments? The clamp study shows a major improvement in glucose uptake in the adipose tissue. Can the authors quantitatively compare the glucose uptake differences when they re-expressed PTPRg in the liver? Have the authors examined the inflammatory status in the adipose tissue? These questions are raised to stimulate the authors to consider the possibility that while important, perhaps even major, there may be other sites, which also contribute to the deficiency phenotype. In particular, adipose tissue (and adipocytes) may deserve the attention of the investigators.”

We thank this Reviewer for suggesting us to extend our detailed analyses to extra-hepatic tissues, including the adipose tissue. To directly test the relevance of PTPR γ in adipocytes (and other tissues) on glucose and insulin homeostasis studies in mice lacking PTPR γ in a tissue-specific manner are needed. However, Cre-conditional *Ptprg* null mice are unavailable. Hence, experiments in mice lacking *Ptprg* in a tissue-specific manner are (currently) technically impossible.

Please, see answer to Reviewer #1 comment #1.

2) “While the NF-kB mediated regulation is sufficiently demonstrated in cells, the same is not quite the case for SIRT1. Also, whether these pathways do act in vivo in a manner similar to cells is examined in a rather limited and indirect manner. It is difficult to make strong conclusions based on the use of resveratrol or nicotinamide riboside. It would be desirable to build stronger links between observation in cells and that of tissues in vivo. Alternatively, the author may want to generate additional insight about the mechanism by which PTPRg impairs metabolic function in vivo. Addressing one of these aspects would have a strong impact on the conclusions of the paper.”

We thank this Reviewer for these suggestions. First, we agree with this Reviewer that while our *in vitro* data are supportive additional *in vivo* studies are required to make a strong statement on the regulatory role of SIRT1 and NFkB on PTPR γ expression. In the revised manuscript we mentioned this limitation of our current findings and highlighted the need to perform future *in vivo* studies. However, we also added new *in vivo* data suggesting that SIRT1 regulates PTPR γ expression (Supplementary Fig 6i)

Pertinent to the mechanistic insight issue; in the last few months we also wanted to address this point experimentally. To this end, we performed new experiments whose results can be found in Supplementary Fig.3f,g,h; Supplementary Fig. 4a; Supplementary Fig. 5 b,c. Based on these new data and our previous findings we have drawn the

conclusion that PTPR γ promotes hepatic glucose production in part by negatively regulating insulin signaling in hepatocytes.

Furthermore, to gain more mechanistic insights into the function of PTPR- γ , we measured the activity of phosphatases/kinases in lysates obtained from liver samples of mice lacking PTPR- γ and their controls. By using arrays with immobilized tyrosine peptide substrates we were able to identify known (e.g. JAK2) and putative new targets of PTPR- γ . Our new results shown in Fig. 2i and Supplementary Table 1 combined with our previous data shown in Fig. 2g,h and Supplementary Fig. 3j indicate that PTPR- γ affects hepatic insulin signaling and glucose metabolism by altering phosphorylation status of proteins whose metabolic role is known (e.g. IR, P85A, AKT) and also of others whose function on hepatic insulin signaling and glucose metabolism is inadequately understood (e.g. LAT, PGFRG, EPHB1, ANXA1, FGFR3, JAK2, FES, VGFR2, LCK, and JAK1). Together, our data demonstrate that lack of PTPR- γ greatly improves glucose homeostasis at least in part by enhancing hepatic insulin sensitivity.

In the revised manuscript we discussed these new results and highlighted the need to perform future studies aimed at directly testing the relevance of some new genes on glucose/insulin homeostasis.

Minor points:

1) “The authors may want to use a more broad and balanced presentation of the field, especially in their introduction and abstract and add some more to their discussion to position their findings and address some of the caveats. For example: Line 26: “Yet, the inflammatory effector(s) linking obesity to insulin resistance is unknown”. This statement is way too strong. Many things are known so stating that it is unknown is not appropriate. Also, it is highly unlikely that there will be one molecular mediator, including PTPR γ , to justify the use of “the mediator” terminology.”

We followed this Reviewer’s suggestions and changed the narrative accordingly.

2) “Line 44-46: “However, pharmacological approaches aimed at targeting the aforementioned mechanisms have had inadequate outcomes or still need to be translated into the clinic 8-14”. Here the authors should represent the literature in a more balanced manner. First, there is now plenty of evidence coming from the humans. There are recent and comprehensive reviews, for example one just published in Nature (PMID: 28179656), to find all relevant references, including those with IL1B blockade, TNF antagonists, CD44, lipid mediators, resolvins, atypical erythropoietin receptor targeting peptide, and others. Second, there are emerging and new strategies to combat inflammation in obesity which has provided exciting benefits against insulin resistance and diabetes in proof-of-principle studies, for example amlexanox from Sautel’s group (PMID:28683283) and immune cell-mediated strategies (PMID:28685960). I would suggest this sentence to be revised into something like “While pharmacological approaches aimed at targeting the aforementioned mechanisms have had encouraging outcomes there is still an unmet need for new, broad and effective tools for translation into the clinic”.”

We thank this Reviewer for all of these suggestions which we found very useful. Thus, we modified references and narrative accordingly.

3) “I would also advise revising the reference list used here. For example, refs 11-14 are all on salsalate. Why not to use one primary paper and perhaps one review and expand the other exciting targets that are being used or tested in humans in this reference list.”

We modified references accordingly.

4) “The authors also have presented the NFkB pathway as if it is the only signaling pathway relevant to obesity-induced inflammation. While this is a very critical mechanism, there are several other key mediators, such as JNK and p38 as well as inflammasome which are not measured or mentioned.”

We totally agree with this suggestion and made reference to JNK, p38, and inflammasome in the Introduction section.

5) “In general a more balanced tone and citations and some attention to the shortcomings of the current study would be a prudent and would not diminish the value of the current study.”

We followed all of this Reviewer’s suggestions. We believe that the revised narrative is more balanced, makes reference to important information that was previously missing, mentions the limitations of the current findings, and highlights new questions that will need to be addressed in future studies. We thank this Reviewer for all of his/her inputs which we believe were instrumental for improving our manuscript.

References:

- Barr, A.J., Ugochukwu, E., Lee, W.H., King, O.N., Filippakopoulos, P., Alfano, I., Savitsky, P., Burgess-Brown, N.A., Muller, S., and Knapp, S. (2009). Large-scale structural analysis of the classical human protein tyrosine phosphatome. *Cell* *136*, 352-363.
- Kianfar, P., Abolfathi, N., and Karimi, N.Z. (2017). Investigating the effect of different transducer stiffness values on the contactin complex detachment by steered molecular dynamics. *Journal of molecular graphics & modelling* *75*, 340-346.
- Miranda, M., Toffali, L., Montresor, A., Scardoni, G., Sorio, C., and Laudanna, C. (2015). Protein Tyrosine Phosphatase Receptor Type gamma Is a JAK Phosphatase and Negatively Regulates Leukocyte Integrin Activation. *Journal of immunology* *194*, 2168-2179.
- Nikolaïenko, R.M., Hammel, M., Dubreuil, V., Zalmai, R., Hall, D.R., Mehzabeen, N., Karuppan, S.J., Harroch, S., Stella, S.L., and Bouyain, S. (2016). Structural Basis for Interactions Between Contactin Family Members and Protein-tyrosine Phosphatase

Receptor Type G in Neural Tissues. *The Journal of biological chemistry* 291, 21335-21349.

REVIEWERS' COMMENTS:

Reviewer #1 (Remarks to the Author):

No further comments

Reviewer #2 (Remarks to the Author):

The authors performed additional experiments and provided more convincing data by addressing the comments that were raised in the previous review. However, there are still a remaining concern.

1. The authors describe that "Combined with our previous data, our new results demonstrate that PTPR- γ is a negative regulator of hepatic insulin signaling in both physiological and obesity/inflammation contexts" in response to my comment. They should also describe its negative regulator role in hepatic insulin signaling in the physiological context in the discussion section.

Reviewer #3 (Remarks to the Author):

In this revised manuscript the authors have satisfactorily addressed my original comments. The manuscript is much improved.

Reviewer #4 (Remarks to the Author):

In their revised manuscript, the authors have provided a robust response and several lines of additional experimental data that advanced the observations and the strength of the conclusions. The authors have also introduced editorial improvements. I have no further comments.

Point-by-point response to remain reviewer's concerns:

Reviewer #2 (Remarks to the Author):

The authors performed additional experiments and provided more convincing data by addressing the comments that were raised in the previous review. However, there are still a remaining concern.

1. The authors describe that “Combined with our previous data, our new results demonstrate that PTPR- γ is a negative regulator of hepatic insulin signaling in both physiological and obesity/inflammation contexts” in response to my comment. They should also describe its negative regulator role in hepatic insulin signaling in the physiological context in the discussion section.

We followed this suggestion and revised the discussion section accordingly.